# Backdoor Vectors: a Task Arithmetic View on Backdoor Attacks and Defenses

## Abstract

Model merging (MM) recently emerged as an effective method for combining large deep learning models. However, it poses significant security risks. Recent research shows that it is highly susceptible to backdoor attacks, which introduce a hidden trigger into a single fine-tuned model instance that allows the adversary to control the output of the final merged model at inference time. In this work, we propose a simple framework for understanding backdoor attacks by treating the attack itself as a task vector. *Backdoor Vector (BV)* is calculated as the difference between the weights of a fine-tuned backdoored model and fine-tuned clean model. BVs reveal new insights into attacks understanding and a more effective framework to measure their similarity and transferability. Furthermore, we propose a novel method that enhances backdoor resilience through merging dubbed *Sparse Backdoor Vector (SBV)* that combines multiple attacks into a single one. We identify the core vulnerability behind backdoor threats in MM: *inherent triggers* that exploit adversarial weaknesses in the base model. To counter this, we propose *Injection BV Subtraction (IBVS)* – an assumption-free defense against backdoors in MM. Our results show that SBVs surpass prior attacks and is the first method to leverage merging to improve backdoor effectiveness. At the same time, IBVS provides a lightweight, general defense that remains effective even when the backdoor threat is entirely unknown.

## 1 Introduction

Model Merging (MM) is an effective and cost-efficient paradigm for updating large pretrained models via weight-space operations Wortsman et al. (2022); Ilharco et al. (2023a); Yadav et al. (2023); Gargiulo et al. (2025); Marczak et al. (2025). It enables the integration of differently fine-tuned models into a single, more capable one. However, research into the security risks of this increasingly popular approach remains limited Yang et al.; Zhang et al. (2024a).

Backdoor attacks have emerged as a major security concern in recent literature Gu et al. (2017); Li et al. (2021); Liang et al. (2024); Qi et al. (2022); Nguyen & Tran (2020); Zhang et al. (2024b); Abad et al. (2024).They are a class of adversarial techniques that implant malicious, hidden behavior into machine learning models by poisoning training data or manipulating the training procedure. It is hard to detect a model compromised in this manner because the model performs as expected on clean inputs, but produces attacker-specified outputs only when a particular trigger is present. Backdoors pose serious risks, especially when models are trained or fine-tuned with data or checkpoints from untrusted sources. Successful attacks can bypass authentication Guo et al. (2021), allow harmful content to evade moderation filters Zhang et al. (2024a) or expose private user data during downstream deployment Guo et al. (2025).

In this work, we study backdoor attacks in the context of model merging. We introduce *Backdoor Vectors (BVs)* – task vectors that capture the information of specific backdoor attacks. The backdoor vector is computed by subtracting the weights of a clean fine-tuned model from those of a backdoored fine-tuned model trained on the same task (see Figure 1a, Section 3).

We show that modeling backdoor attacks as task vectors using Task Arithmetic (TA) Ilharco et al. (2023b) offers a simple and effective framework to analyze and quantify their behavior. Addition and negation map naturally to attack and defense, while BV analysis enables comparison and reveals

Figure 1: **Backdoor Attack as Task Vector = Backdoor Vector (BV). (a)** A BV is the element-wise difference between backdoored and clean fine-tuned model parameters. **(b)** Adding a BV injects a backdoor; subtracting it weakens the attack. **(c)** Like task analogies, backdoor analogies reveal relationships between attacks. We define *Backdoor Transfer* as positive when Attack Success Rate (ASR) is strengthened and negative when is weakened by another task vector. **(d)** We show a method of merging BVs that yields a significantly stronger attack.

transfer dynamics during MM. We introduce *Sparse Backdoor Vectors (SBVs)*, which merge multiple BVs into a more resilient and potent attack, yielding higher Attack Success Rate (ASR) after merging.

Finally, we identify the root of the backdoor threat in model merging as the white-box access to the pre-trained foundational model, an inherent requirement in all model merging scenarios. Although such access enables seamless collaboration, it also reveals adversarial vulnerabilities within the model, known as *inherent triggers* Tao et al. (2024); Wenger et al. (2022); Tao et al. (2022). We are the first to show that *inherent triggers* have high resilience in the MM process, but also share *inherent* similarities. Building upon this observation, we propose *Injection BV Subtraction (IBVS)* defense method to mitigate backdoor risks for MM.

We summarize the contributions of this work as follows:

- We introduce *Backdoor Vectors (BVs)* and show that viewing backdoor attacks through task arithmetic reveals new insights, such as an intuitive understanding of the interplay between backdoor attacks and defenses, enabling a more effective framework for measuring their similarity, transferability, and resilience in model merging.
- We explore backdoor attacks as task vectors and show that merging multiple Backdoor Vectors produces a single, stronger and more resilient *Sparse Backdoor Vector (SBV)*. Our simple BV merging strategies outperform state-of-the-art attacks in effectiveness and in merging resilience.
- We identify the core vulnerability enabling backdoor threats in model merging as the inherent white-box access to the pre-trained foundational model. To address this, we propose *Injection BV Subtraction* – a defense method to mitigate backdoor risks even when the backdoor threat is entirely unknown.

## 2 PROBLEM SETTING

**Model Merging**  We follow the standard procedure of merging vision models Yadav et al. (2023); Ilharco et al. (2022); Tang et al. (2023); Ortiz-Jimenez et al. (2024); Yang et al. (2024; 2023). We use CLIP model Radford et al. (2021) $M = \{V, T\}$ with an image encoder $V$ and a text encoder $T$. CLIP aligns image and text embeddings in a joint space enabling zero-shot classification by comparing an image to textual class descriptions via cosine similarity. The predicted label corresponds to the class with the highest similarity score. To improve performance on specific tasks, a common approach is to fine-tune only the vision encoder $V$ while freezing the text encoder $T$ (see Appendix A.2 for details).

Let $M_{\boldsymbol{\theta}}$ denote a CLIP-like model parameterized by weights $\boldsymbol{\theta}$, and let $V_{\theta}$ represent its visual encoder. We denote the weights of the base pre-trained model $M_{\text{pre}}$ by $\theta_{\text{pre}}$, and the weights of a model fine-tuned on dataset $\mathcal{D}^{(t)}$ from task $t$ as $\theta^{(t)}$. The corresponding *task vector* is defined as the element-wise difference between the fine-tuned and pre-trained weights:

$$\Delta\boldsymbol{\theta}^{(t)} = \boldsymbol{\theta}^{(t)} - \boldsymbol{\theta}_{\text{pre}}. \tag{1}$$

Suppose that we are given $n$ such task vectors $\{\Delta\boldsymbol{\theta}^{(1)}, \ldots, \Delta\boldsymbol{\theta}^{(n)}\}$, obtained from different fine-tuning instances (potentially across different tasks or training configurations). The model merging

process aims to construct a new set of weights $\boldsymbol{\theta}_{\text{merged}}$ by combining these task vectors into a merged task vector $\Delta\boldsymbol{\theta}_{\text{merged}}$, which is then added to the pre-trained weights:

$$\boldsymbol{\theta}_{\text{merged}} = \boldsymbol{\theta}_{\text{pre}} + \lambda\Delta\boldsymbol{\theta}_{\text{merged}}. \tag{2}$$

where $\lambda$ is a scaling factor determined on a held-out validation set. The exact strategy for computing $\Delta\boldsymbol{\theta}_{\text{merged}}$ may vary, including simple averaging (Wortsman et al., 2022), heuristics aiming at reducing interference between models Yadav et al. (2023); Yu et al. (2024); Marczak et al. (2024); Wang et al. (2024a; 2025), or more sophisticated optimization-based approaches Yang et al. (2024; 2023).

We consider two distinct merging scenarios: *single-task* and *multi-task*. In the former, all merged models are fine-tuned on the same downstream task, differing only in training seeds, augmentations, or other hyperparameters. This setup is commonly used for robustness or performance improvement on a single objective, as shown in Model Soups Wortsman et al. (2022). *Multi-task* scenario follows the setting from learning via addition from Task Arithmetic Ilharco et al. (2023a): given models fine-tuned on different tasks, we aim to fuse them to obtain a model capable of multi-task generalization. In this work, we focus primarily on the *single-task* merging scenario, since it is harder to create a resilient backdoor in this setup. We aggregate multiple independently fine-tuned models $M^{(\text{t})}$, for $t = 1, \ldots, n$, all originating from a common pre-trained model $M_{\text{pre}}$. Although our emphasis is on the *single-task* case, we note that similar attack strategies may seamlessly extend to the *multi-task* setting.

**Threat Model in Model Merging.**  We adopt the threat model from Zhang et al. (2024a) and extend it to include the defender's perspective.

*Attack scenario.* We consider a threat model in which the adversary acts as a model provider. The adversary publicly releases a $M_{\text{backdoored}}$ (fine-tuned for a task denoted as *adversary task*) with competitive utility to increase its chances of inclusion in a merged model. When $M_{\text{backdoored}}$ is incorporated into the model merging process, $M_{\text{merged}}$ behaves according to the adversary's intent, which can lead to major security breaches, as discussed in Section 1. We assume that only one of the models originates from the adversary, while the rest are contributed by benign providers. We refer to any task other than the adversary's as a *clean task*.

*Adversary's Knowledge.* The adversary has a dataset $\mathcal{D}_{\text{adv}}$ corresponding to a single task. Similarly to clean model contributors, adversary has white-box access to $M_{\text{pre}}$, since they fine-tune a base model $M_{\text{pre}}$ to obtain $M_{\text{backdoored}}$. We assume that the adversary contributes only one model and has no information about other models, merging algorithms, or coefficients used in the merging process.

*Defense scenario.* The defender's objective is to construct a reliable and secure merged model from independently submitted components, ensuring high utility on the target task while preventing the inclusion of any malicious behavior. Given that individual models may originate from untrusted sources, the defender aims to preserve the functional performance of the merged model on *clean tasks*, while mitigating the risk of backdoor activation.

*Defender's Knowledge.* We assume that the defender lacks prior knowledge regarding whether any of the models have been backdoored. Furthermore, the defender has no access to the internal training data or methodologies employed by the individual contributors. Although the merging algorithm and its associated coefficients are under the control of the defender, the presence and nature of any injected trigger patterns remain unknown. Consequently, the defender must rely exclusively on the behavior of the submitted models to maintain the security and reliability of the merged model.

## 2.1 BACKDOOR ATTACK

Let $x$ denote a clean input image and define the trigger as $t = \{m, \delta\}$, where $m \in \{0, 1\}^{H \times W}$ is a binary mask specifying the location of the trigger, and $\delta \in \mathbb{R}^{H \times W \times C}$ encodes the trigger pattern. The poisoned image is constructed by an injection function $x \circledast t$, defined as:

$$x \circledast t = \delta \odot m + (1 - m) \odot x, \tag{3}$$

where $\odot$ denotes element-wise (pixel-wise) multiplication. The objective of the backdoor attack is to train a model so that $x$ is classified correctly, that is, $f(x) = y$, but $f(x \circledast t) = c$, where $c$ is an attacker-specified target class. In this work, we call $t$ *inherent (★)* if the trigger patch was optimized to backdoor $M_{\text{pre}}$ (e.g., by adversarial attack using white-box access to $\theta_{\text{pre}}$) and *injected (⊕)* if the trigger is a fixed pattern injected only to $M_{\text{backdoored}}$.

Table 1: **Sparsity H(x) of TVs, BVs, and SBVs**: $x \in (0, 1)$; values near 1 indicate higher sparsity.

|  |  | CIFAR100 | ImageNet100 | TinyImageNet100 |
|---|---|---|---|---|
|  | TV | 0.3270 | 0.3190 | 0.3234 |
| **BadMerging** | $BV$ | 0.3330 | 0.3261 | 0.3273 |
|  | $SBV$ (Ours) | **0.5482** | **0.5407** | **0.5455** |
| **BadNets** | $BV$ | 0.3296 | 0.3238 | 0.3247 |
|  | $SBV$ (Ours) | **0.5256** | **0.5262** | **0.5247** |

Table 2: **TVs Pairwise Cosine Similarity.**

| Task | CIFAR100 | ImageNet100 | TinyImageNet100 |
|---|---|---|---|
| TV | 0.5646 | 0.5421 | 0.5438 |
| BV (BadMerging) | 0.5986 | 0.7238 | 0.6683 |
| BV (BadNets) | 0.3859 | 0.4671 | 0.4555 |
| BV (BadMerging_BadNets) | 0.5026 | 0.6653 | 0.5747 |

---

**Algorithm 1** Sparse Backdoor Vector (SBV)

**Require:** Backdoored task vector $\boldsymbol{\Delta}_{\text{backdoored}}$, clean task vectors $\{\boldsymbol{\Delta}_{\text{clean}}^{(t)}\}_{t=1}^{k}$, sparsification type $st \in \{\text{SC}, \text{RND}\}$
1: **for** each $t = 1$ to $k$ **do**
2:     $\boldsymbol{BV}^{(t)} \leftarrow \boldsymbol{\Delta}_{\text{backdoored}} - \boldsymbol{\Delta}_{\text{clean}}^{(t)}$
3: **end for**
4: $\mathbf{s} \leftarrow \sum_{t=1}^{k} \boldsymbol{BV}^{(t)}$
5: $\boldsymbol{\mu} \leftarrow \texttt{sparse\_mask}(\{\boldsymbol{BV}^{(t)}\}, st)$
6: $\mathbf{SBV} \leftarrow \mathbf{s}_j \odot \mu_j$
7: **return SBV**

---

**Algorithm 2** `sparse_mask` Function

**Require:** Backdoor vectors $\{\boldsymbol{BV}^{(t)}\}_{t=1}^{k}$, type $st \in \{\text{SC}, \text{RND}\}$
**Ensure:** Mask $\boldsymbol{\mu} \in \{0, 1\}^d$
1: $S \leftarrow \texttt{sign}(\{\boldsymbol{BV}^{(t)}\})$
2: $s_0 \leftarrow S[1]$
3: $c \leftarrow \texttt{all}(S == s_0, \text{dim} = 0)$
4: $nz \leftarrow \texttt{all}(S \neq 0, \text{dim} = 0)$
5: $\boldsymbol{\mu} \leftarrow c \wedge nz$
6: **if** $st = \text{RND}$ **then**
7:     $\boldsymbol{\mu} \leftarrow \texttt{shuffle}(\boldsymbol{\mu})$
8: **end if**
9: **return** $\boldsymbol{\mu}$

---

# 3 BACKDOOR VECTORS

**Intuition.** Task vectors Ilharco et al. (2023a) provide a simple way of thinking about modifying the capabilities of a model: adding a task vector improves the performance on the corresponding task while subtracting it enables unlearning of the task. Task vectors are also effective in modifying the properties of models that are not usually considered a task, e.g. the toxicity of text generation Ilharco et al. (2023a). Therefore, we treat model vulnerabilities as any other task and propose to look at backdoor attacks as *backdoor vectors*.

**Definitions.** Following Section 2, let $\theta_{\text{pre}}$ be the weights of a pre-trained model and $\theta^{(t)}$ the weights after fine-tuning on task $t$. The model fine-tuned by a benign provider is optimized solely to improve the performance on task $t$, resulting in weights $\theta_{clean}^{(t)}$. The adversary aims to produce a model that performs well on the task $t$ but also contains certain vulnerabilities (i.e. backdoor) resulting in weights $\theta_{backdoored}^{(t)}$. The backdoor vector $\text{BV}^{(t)}$ is a parameter-wise difference between the backdoored weights $\theta_{backdoored}^{(t)}$ and the clean weights $\theta_{clean}^{(t)}$, i.e. $BV^{(t)} = \theta_{backdoored}^{(t)} - \theta_{clean}^{(t)}$. We use a scaling coefficient $\lambda_{BV}$ to modulate the strength of an attack (or a defense). We denote BVs from backdoor attacks with $\oplus$ and $\bigstar$ triggers as $BV_\oplus$ and $BV_\bigstar$, respectively.

*Backdoor Transfer* determines how another task or backdoor vectors $V$ interact with the primary backdoor vector (Figure 1c). Positive transfer occurs when $V$ strengthens the primary backdoor, while negative transfer weakens it and can be used for defense. Notably, this interaction may not be symmetrical: one attack vector can strengthen another, but the reverse might not occur (see Figure 9 in the Appendix).

## 3.1 BV IMPROVEMENT BY MERGING: SPARSE BACKDOOR VECTORS (SBV)

Merging backdoored task with clean tasks can introduce negative backdoor transfer, weakening the attack. To counter this, we introduce BV merging method (Figure 1d) combining multiple backdoor attacks into a stronger one that can withstand model merging. We propose *sparsifying* BVs to retain only the most consistent and influential malicious components. This enhances the resulting sparse BV (SBV), allowing it to reinforce the trigger and persist despite dilution from clean models.

Algorithms 1 and 2 detail our sparsification procedure. We adapt a simple idea used in MM to reduce TV interference for backdoors and propose sign-consistent sparsification ($SBV_{\text{SC}}$), which retains components with consistent weigths signs across merged BVs.

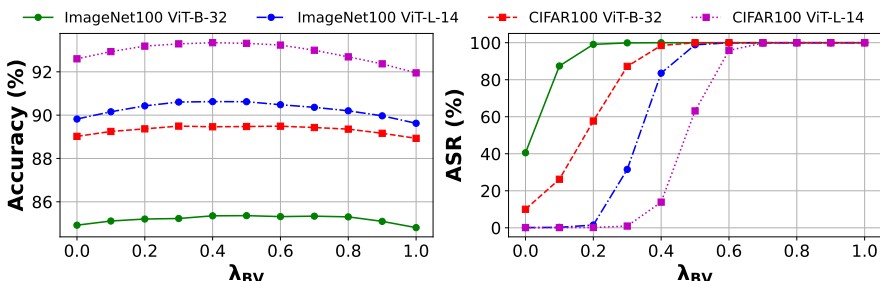

Figure 2: $\theta_{\text{clean}} + \lambda BV = \theta_{\text{backdoored}}$. Increasing the $\lambda$ of BV added to $\theta_{\text{clean}}$ preserves accuracy (left) but sharply raises attack success rate (right).

### 3.2 BV USED FOR DEFENSE: INJECTION BV SUBTRACTION (IBVS)

We show that state-of-the-art backdoor attacks on MM rely on *inherent triggers* that produce highly similar and aligned $BV_\bigstar$ vectors, enabling strong cross-attack transfer and revealing shared, general-izable structure of backdoors (see Section 4.4). To exploit this, we propose *Injection BV Subtraction (IBVS)* – a defense that subtracts a fixed $BV_\oplus$ (e.g., BV created from a simple white square trigger; see Figure 6) to suppress unknown $\bigstar$ attacks. **IBVS** requires no knowledge of the adversary's dataset, labels, target class, or trigger. The defender only needs to train a fixed $\oplus$ trigger on any dataset, compute $BV_\oplus$, and subtract it – using task arithmetic as a simple yet effective tool to mitigate backdoor influence.

## 4 EXPERIMENTS

### 4.1 EXPERIMENTAL SETUP

We follow the common model merging experimental setup for backdoor attacks introduced by Zhang et al. (2024a). We present more detailed information about the details in the Section B and present additional results in the Section C, including results for multi-task setup, different architectures (ConvNext and ViT-L-14) and different merging types.

**Metrics.** *Clean Accuracy (CA)* is the accuracy on clean test data obtained by a clean merged model (all provided task vectors are clean). *Backdoored Accuracy (BA)* is the accuracy on clean test data obtained by a backdoored merged model (one provided task vector – adversary task – is backdoored). *Attack Success Rate (ASR)* is a fraction of triggered test images from the adversary task that are predicted as the target class by $M_{merged}$. An attack is considered successful when the ASR is high and the backdoored model performs similarly well as clean model (BA $\approx$ CA). We use *Hoyer sparsity* Hoyer (2004) to compare the sparsity of $TVs$, $BVs$ and $SBVs$.

**Datasets.** We conduct our experiments on the following datasets. We use CIFAR100 Krizhevsky & Hinton (2009) and ImageNet100 Deng et al. (2009) as adversary tasks. In single-task setup, we merge one adversary task with the rest clean using the same dataset. In multi-task setup, folllowing Zhang et al. (2024a), we merge adversary task with five clean tasks: Cars Krause et al. (2013), SUN397 Xiao et al. (2010), EuroSAT Helber et al. (2019), GTSRB Stallkamp et al. (2011) and Pets Parkhi et al. (2012).

**MM.** We use TA Ilharco et al. (2023a) as MM algorithm. We use ViT-B-32, ViT-L-14 and ConvNext as visual encoders of CLIP models, the first being the default for the experiments in the main part of our work. For BV merging we use weights averaging as baseline Wortsman et al. (2022) for our proposed $SBV_{SC}$ method.

**Backdoor attacks.** We use the simplest classic BadNets Gu et al. (2017) attack as a representative of fixed injected ($\oplus$) triggers as well as current state-of-the-art attack on MM (using inherent $\bigstar$ triggers): BadMerging Zhang et al. (2024a). We compare these attack types using BVs, assess their transferability in MM, and show that our sparsification method substantially improves trigger resilience and final ASR in both cases. We set the trigger size to be 1% of pixels in the image.

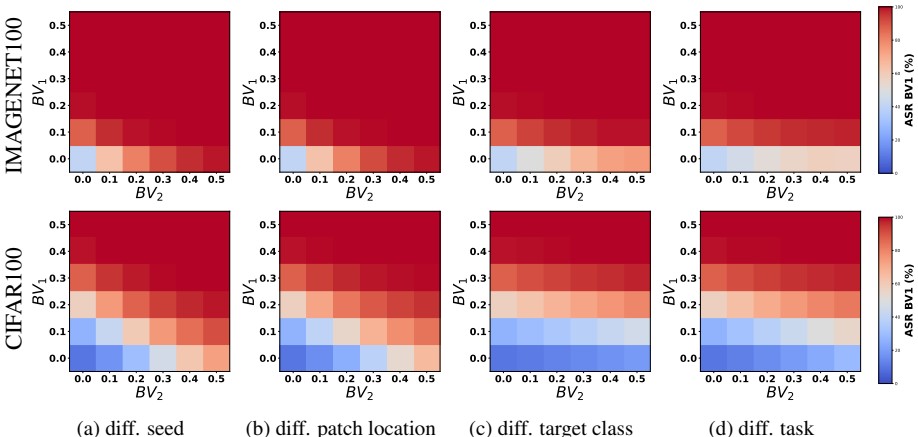

(a) diff. seed      (b) diff. patch location      (c) diff. target class      (d) diff. task

Figure 3: **ViT-B-32: Positive Backdoor Transfer Across Attacks** ($BV_2 \rightarrow BV_1$). Adding $BV_2$ increases the ASR of $BV_1$. Axes show $\lambda_{BV_1}, \lambda_{BV_2}$; (a–c) use BVs from the same task (ImageNet100 or CIFAR100). Strong transfer occurs across seeds **(a)** and patch locations **(b)**, but is weaker across target classes **(c)** and tasks **(d)**.

## 4.2 KEY OBSERVATIONS

Our four key observations align with the concepts illustrated in Figure 1:

**BV addition represents backdoor attack.** ASR grows sharply with increasing $\lambda_{BV}$ (see Figure 2), confirming that the effectiveness of the backdoor attack improves as the scaling factor increases.

**BVs have high transferrability.** Adding a BV from another attack notably boosts ASR of primary attack, as shown in Figure 3. Interestingly, backdoor transfer remains strong across trigger patches with different seeds or locations (Figure 3a,b), suggesting triggers are not fixed to one position and can be reinforced by varying or duplicating placement. Positive transfer across tasks or target classes also occurs, but at a lower magnitude.

**BV merging: proposed SBV enhances attack by merging multiple attacks.** Figures 4 and 5 show ACC/ASR trajectories throughout MM, with the optimal point in the top-right. Naive merging (AVG) offers little gain over a single BV. $SBV$ uses sign consistency to produce more sparse BV (see Table 1) that better preserve backdoor information through merging. In 7 of 8 cases, $SBV_{SC}$ performs best, highlighting that random sparsification ($SBV_{RND}$) is insufficient – sign consistency is the key to maximizing effectiveness. Figure 4 shows that $SBV$ significantly improves backdoor resilience in single-task MM, surpassing the state-of-the-art. We are the first to show a *non-inherent* backdoor attack that withstands the merging process (Figure 5). It further highlights the importance of our backdoor merging method, showing that merging can significantly strengthen even simplest backdoor attack types. As shown in Table 16, our method outperforms both weight-averaging and no-merging approaches.

**BV subtraction as backdoor defense.** Figure 6 shows how defenders can leverage backdoor transfer to improve MM robustness. Our proposed IBVS reduces ASR with minimal accuracy loss (Table 4) requiring only a fixed $\oplus$ trigger trained on any dataset to compute and subtract $BV_\oplus$ during merging.

## 4.3 ADDITIONAL INSIGHTS

**BV addition does not degrade the accuracy.** The results in Figure 2 illustrate the effects of increasing the scaling coefficient of the BV ($\lambda_{BV}$) on the attack strength. The accuracy remains stable as the $\lambda_{BV}$ increases, indicating that BV does not degrade the clean performance of the model.

**Reducing ASR via $BV_\square$ Subtraction.** Subtracting $\mathbf{BV_\square}$, derived from an attack with a fixed white square trigger, lowers the ASR of unknown $\bigstar$ attacks with minimal accuracy loss. This forms the basis of our **IBVS** defense, which builds a fixed $\oplus$-trigger BV and subtracts it during model

Table 3: **ViT-B-32: Backdoor Merging: Single-Task Attack Results.** We merge $1 \times \theta_{\text{backdoored}} + 9 \times \theta_{\text{clean}}$ with $\lambda = 0.1$, reporting mean ± std. dev. **BV merging:** $AVG$ — weight averaging **Methods:** BN($\oplus$) — BadNets, BM($\bigstar$) — BadMerging. ViT-B-32.

| Backdoor Attack | | | Adversary task: CIFAR100 | | | Adversary task: IMAGENET100 | | |
|---|---|---|---|---|---|---|---|---|
| Setting | Method | BV merging | CA | BA | ASR | CA | BA | ASR |
| Single-task | BN($\oplus$) | - | 89.70 ± 0.03 | 89.73 ± 0.03 | 0.2 ± 0.12 | 85.88 ± 0.07 | 85.88 ± 0.04 | 0.26 ± 0.1 |
| | BN($\oplus$) | AVG | 89.70 ± 0.03 | **89.75 ± 0.03** | 0.21 ± 0.12 | 85.88 ± 0.07 | **86.0 ± 0.03** | 0.27 ± 0.1 |
| | BN($\oplus$) | $SBV_{RDM}$ (Ours) | 89.70 ± 0.03 | 89.72 ± 0.04 | 0.23 ± 0.12 | 85.88 ± 0.07 | **86.0 ± 0.06** | 0.33 ± 0.14 |
| | BN($\oplus$) | $SBV_{SC}$ (Ours) | 89.70 ± 0.03 | 89.73 ± 0.12 | **12.23 ± 10.52** | 85.88 ± 0.07 | 85.8 ± 0.06 | **29.45 ± 12.36** |
| | BM($\bigstar$) | - | 89.70 ± 0.03 | **89.74 ± 0.04** | 27.29 ± 12.14 | 85.88 ± 0.07 | 85.91 ± 0.04 | 84.01 ± 7.83 |
| | BM($\bigstar$) | AVG | 89.70 ± 0.03 | 89.72 ± 0.03 | 27.56 ± 12.17 | 85.88 ± 0.07 | 85.96 ± 0.02 | 84.79 ± 7.48 |
| | BM($\bigstar$) | $SBV_{RDM}$ (Ours) | 89.70 ± 0.03 | 89.72 ± 0.01 | 35.93 ± 13.64 | 85.88 ± 0.07 | **85.97 ± 0.04** | 92.55 ± 3.83 |
| | BM($\bigstar$) | $SBV_{SC}$ (Ours) | 89.70 ± 0.03 | 89.69 ± 0.06 | **97.21 ± 2.3** | 85.88 ± 0.07 | 85.83 ± 0.09 | **99.99 ± 0.01** |

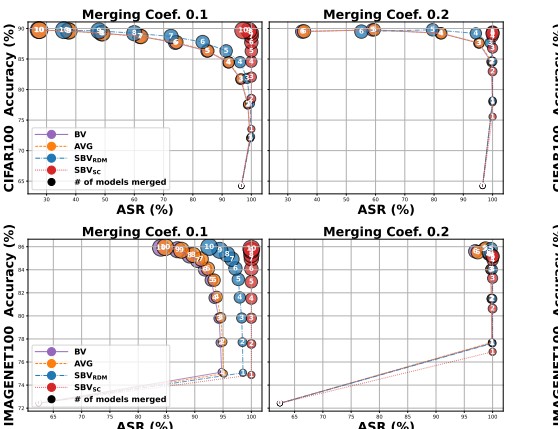

Figure 4: **ViT-B-32: ACC/ASR trajectories for inherent ($\bigstar$) triggers in first-backdoored-rest-clean single-task MM.**

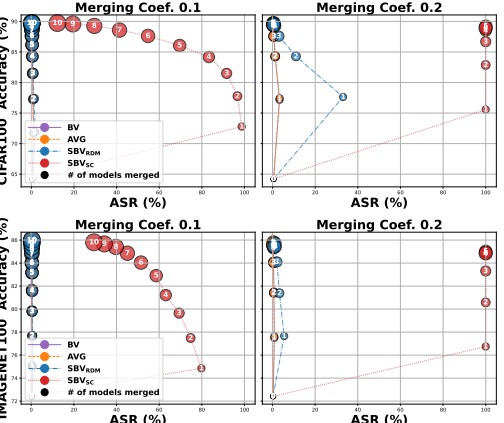

Figure 5: **ViT-B-32: ACC/ASR trajectories for injected ($\oplus$) triggers in first-backdoored-rest-clean single-task MM.**

merging to weaken state-of-the-art MM attacks (see Table 4). For more in-depth experiments on IBVS performance see Section C.1.2 in the appendix.

**Backdoor transfer is asymmetric.** Adding $BV_\oplus$ with small $\lambda_{BV_\oplus}$ increases the ASR for $BV_\bigstar$ attack, but the reverse does not occur (see Figure 9 in the appendix). Similarly, smaller triggers enhance larger ones more than the reverse.

## 4.4 ADDITIONAL OBSERVATIONS ON INHERENT VS. INJECTED TRIGGERS IN MM

**Inherent similarity of backdoor attacks with $\bigstar$ triggers.** State-of-the-art backdoor attacks on MM Zhang et al. (2024a) exploit adversarial vulnerabilities in $M_{\text{pre}}$ (see Figure 7 and Figure 8). As a result, $M_{\text{pre}}$-optimized attacks (attacks with $\bigstar$ triggers) are *inherently* similar, yielding highly aligned

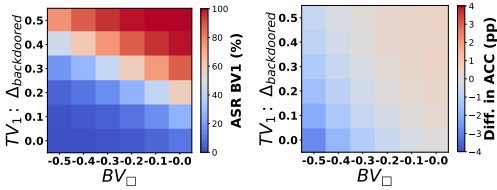

Figure 6: **Vit-B-32: Injection BV Subtraction (IBVS) Defense**: subtracting the BV of a white-square ($\square$) trigger. *Left:* Removing $BV_\square$ lowers the ASR of an unknown attack ($BV_1$), highlighting cross-type transferability (see Figure 9). *Right:* This ASR drop slightly reduces clean accuracy (vs. (0.0, 0.0)) but can be mitigated by natural addition of clean task vectors in MM process.

Table 4: **ViT-B-32 – Backdoor Defense: Injected BV Subtraction (IBVS).** Single-task defense results for merging $1 \times \theta_{\text{backdoored}} + 9 \times \theta_{\text{clean}}$ with $\lambda = 0.1$, reporting mean BA and ASR. BV merging: $AVG$ — weight averaging. IBVS: $IBVS_\square$ uses a fixed white-square trigger; $IBVS_{BN}$ uses a wavelet trigger from BadNets. Backbone: ViT-B-32. No defense: Badmerging attack.

| Setting | Adversary task: CIFAR100 | | | | | | | | Adversary task: ImageNet100 | | | | | | | |
|---|---|---|---|---|---|---|---|---|---|---|---|---|---|---|---|---|
| | BV | | AVG | | $SBV_{RND}$ | | $SBV_{SC}$ | | BV | | AVG | | $SBV_{RND}$ | | $SBV_{SC}$ | |
| | BA | ASR | BA | ASR | BA | ASR | BA | ASR | BA | ASR | BA | ASR | BA | ASR | BA | ASR |
| No defense | 89.74 | 27.29 | 89.72 | 27.56 | 89.72 | 35.93 | 89.69 | 97.21 | 85.91 | 84.01 | 85.96 | 84.79 | 85.97 | 92.55 | 85.83 | 99.99 |
| IBVS$_\square$ (Ours) | 89.58 | 20.65 | 89.36 | 23.59 | 89.41 | 30.81 | 89.61 | 94.69 | 85.53 | 66.83 | 85.13 | 81.22 | 85.15 | **89.74** | 85.47 | **99.83** |
| IBVS$_{BN}$ (Ours) | 89.57 | **16.56** | 89.41 | **19.36** | 89.42 | **25.37** | 89.46 | **89.21** | 85.61 | **65.26** | 85.14 | **80.89** | 85.14 | 90.17 | 85.59 | 99.89 |

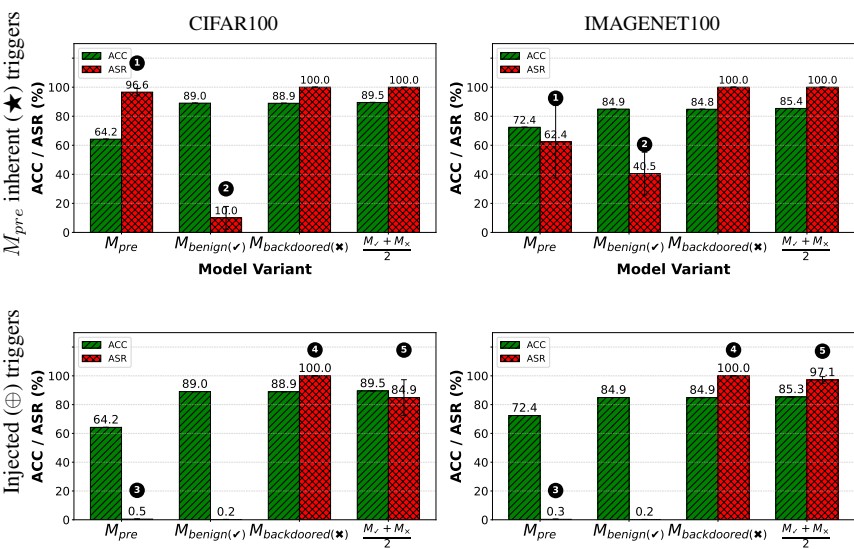

Figure 7: **Comparison of inherent (★) and injected (⊕) backdoor triggers in model merging.** State-of-the-art attacks on MM use ★ triggers – adversarial vulnerabilities of the base model $M_{\text{pre}}$. They ❶ directly affect $M_{\text{pre}}$ and are effective even before merging with $M_{\text{backdoored}}$ and ❷ remain effective after merging with clean (benign) model. In contrast, ⊕ triggers, used in classical backdoor attacks, rely on fixed patches, ❸, ❹ require merging with $M_{\text{backdoored}}$ to become effective, and ❺ degrade faster after merging with clean models.

BVs with strong cosine similarity (see Table 2). Interestingly, $BV_\bigstar$ exhibit greater mutual similarity than TVs from the same task. This suggests that inherent triggers share generalizable structure, as evidenced by high $BV_\bigstar$ similarity and strong backdoor transfer across attacks (see Figure 3).

**Similarity and transfer of $BV_\bigstar$ vs $BV_\oplus$.** Interestingly, BVs from different attacks can be more similar than $BV_\oplus$s or same-task TVs. Additionally, as shown in Figure 9, backdoor transfer is notably stronger from $BV_\oplus$ to $BV_\bigstar$, reinforcing the idea that many triggers share a generalizable structure – an insight central to our IBVS defense.

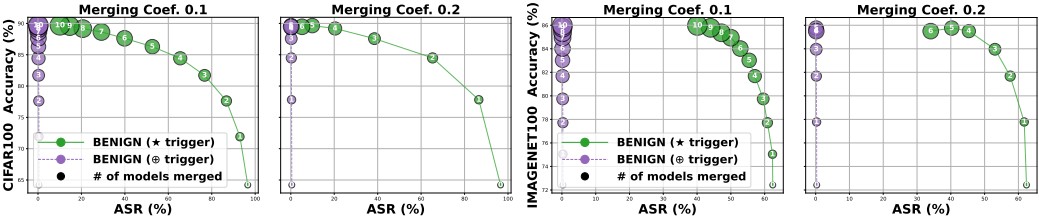

Figure 8: **ACC/ASR trajectories for inherent (★) and injected (⊕) triggers in all-clean single-task MM.** We demonstrate that ★ triggers exploit the adversarial vulnerabilities of the base model $M_{\text{pre}}$, maintaining their effectiveness even after merging multiple clean (benign) models. Conversely, ⊕ triggers are completely ineffective in the absence of $M_{\text{backdoored}}$ model.

## 5 DISCUSSION AND RELATED WORK

We provide an extended discussion of related work in Appendix A.1 and concentrate here on recently proposed backdoor attacks on MM to better contextualize our contributions.

**Backdoor Attacks on Model Merging.** Model merging usually operates under the assumption that task vectors are obtained from reliable, trustworthy sources and, therefore, can be utilized without posing security risks. However, recent studies Zhang et al. (2024a); Hsu et al. (2025); Yin et al. (2024); Guo et al. (2025); Hammoud et al. (2024) have raised concerns about the security of the model merging process, showing that maliciously crafted task vectors can potentially compromise the integrity of the merged model. A key example is BadMerging Zhang et al. (2024a), which demonstrates that an adversary can implant backdoors into a model by crafting task vectors that, when merged, result in malicious model behavior. Follow-up work has expanded this threat model. BADTV Hsu et al. (2025), shows how third-party task vectors can to insert hidden functionality into the merged model without access to training data. Other techniques, like LoBAM Yin et al. (2024), take advantage of low-rank adaptation modules to inject backdoors in a parameter-efficient manner, preserving standard performance on clean data. Recent works explore privacy leakage Guo et al. (2025) and multi-stage attacks Guo et al. (2025); Lu et al. (2025), where backdoors are composed or activated through a sequence of merging steps.

We argue that analyzing backdoor attacks on MM is a complex issue and standard BA and ASR metrics used in the above works to describe backdoor attack performance is not detailed enough to fully quantify their behavior. We propose Backdoor Vector framework for the emerging field of backdoor attacks on MM for better understanding of backdoors. We think it is a crucial step to mitigate emerging backdoor threats. Our work is a substantial step forward in this emerging field, as we present unified framework as well as its direct practice benefits, like SBV method that outperform current state-of-the-art by a large margin, with no significant computations added.

**Defenses.** In response, several defences have been proposed to counteract backdoors in model merging. Subspace masking Yang et al. (2025) introduced by Yang et al. attempts to constrain the merged model weights to parameter regions that are less susceptible to adversarial attacks. Another, proposed by Arora et al. Arora et al. (2024), makes the counterintuitive observation that merging multiple backdoored models may in some cases cancel out their respective backdoors – a phenomenon they term the "Free Lunch" effect. Chen et al. Chen et al. (2024) pursue an active approach in identifying conflicts between task vectors and leverage these oppositions to neutralize backdoor effects.

Our results put the "Free Lunch" hypothesis under the new light and show that the adversary *can* craft much more resilient backdoors with *inherent triggers*, and also significantly boost the standard backdoor merging resilience by merging multiple backdoors into one using SBV method. Additionally, those insights lead us to formulate the hypothesis that backdoor attacks share *inherent* similarities, which can be also used for defense (like our proposed IBVS).

## 6 CONCLUSIONS

In this work, we introduced a novel perspective on backdoor attacks in model merging by framing them as task arithmetic problems. By defining and analyzing *Backdoor Vectors* (BVs), we provided a simple yet powerful abstraction that enables the injection, transfer, and mitigation of backdoors via vector operations. We show that BVs expose key insights into attack dynamics and enable stronger attacks through *Sparse Backdoor Vectors* (SBVs), which merge multiple BVs into a single, resilient and highly effective threat. On the defense side, we proposed *Injection BV Subtraction (IBVS)* – a lightweight, assumption-free method for mitigating unknown backdoor attacks. Our framework unifies multiple phenomena observed in model merging security and opens new directions to understand and secure model merging pipelines.

**Limitations** Our work shares a common assumption in the MM literature that task vectors are linearly additive. This assumption holds when operating in a linear connectivity regime, which is induced by the scale of the model and amount of pretraining. This work focuses on CLIP-like vision models and image classification tasks; its applicability to other architectures (e.g., large language models) remains to be explored.

### 6.1 ETHICS STATEMENT

Recently, backdoor attacks and (more broadly) data poisoning attacks have gained increasing attention due to their significance as a threat to open model sharing paradigms. By sharing results of our investigations on backdoors in MM we aim to broaden the access to the information on such threats in the community and increase the interest of researchers in active research to counter newly discovered attacks.

Our work on understanding backdoor attacks in model merging enables defenders to better detect, understand, and counteract such threats. By shedding light on the attacks using task arithmetic, we aim to inform the research community and practitioners deploying merged models – do not underestimate the data poisoning threat in MM paradigm.

Framing backdoor attacks as TVs provides powerful tools to study them in MM beyond standard CA/BA/ASR metrics (similarity, backdoor transfer, 'the shared internal structure' of inherent triggers), which is beneficial for the community.

Additionally, as most of backdoor attacks, our SBVs can be used for IP protection by allowing a model provider to embed a backdoor as a watermark, enabling verification of their model's presence even after it has been merged with others.

### 6.2 REPRODUCIBILITY STATEMENT

For reproduction of our experiments see Section 4.1 in the main section of our work. All remaining details are described in Section B in the Appendix.

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

# A   ADDITIONAL INFORMATION

## A.1   RELATED WORK ON MODEL MERGING AND BACKDOOR ATTACKS

**Model Merging**    Model merging aims to integrate weights of multiple models. Historically, averaging weights along the training trajectory was used to improve the generalization of the model Izmailov et al. (2018). More recently, many works (Wortsman et al., 2022; Rame et al., 2022; Jang et al., 2024) have shown that merging multiple independent fine-tunings on a given task improves model's performance in-distribution and out-of-distribution. On the other hand, model merging can be used to integrate the knowledge of multiple models fine-tuned on disjoint tasks to obtain a multi-task models Yadav et al. (2023); Ilharco et al. (2022); Tang et al. (2023); Ortiz-Jimenez et al. (2024); Yang et al. (2024; 2023).

**Backdoor Attacks**    Backdoor attacks Gu et al. (2017); Liu et al. (2018) are a class of adversarial techniques that implant malicious hidden behavior into machine learning models by poisoning training data Chen et al. (2017); Turner et al. (2019); Zhang et al. (2024b); Tran et al. (2024) or manipulating the training procedure Bagdasaryan et al. (2020); Goldblum et al. (2022); Huang et al. (2024); Wang et al. (2024b). It is hard to detect a model compromised in this manner because the model performs as expected on clean inputs, but produces attacker-specified outputs only when a particular trigger is present. Recent works Tao et al. (2024); Wenger et al. (2022); Tao et al. (2022); Wang et al. (2022) show that pre-trained models can also exhibit *inherent triggers*—natural adversarial vulnerabilities that act as backdoors without explicit trigger injection.

## A.2   CLIP-LIKE CLASSIFIERS

Contrastive Language–Image Pretraining Radford et al. (2021) (CLIP)-like classifier employ contrastive learning to align visual and textual modalities within a joint embedding space. The classifier model $M = \{V, T\}$ uses an image encoder $V(x)$ and a text encoder $T(t)$, respectively. Given an image $x$ and a set of textual class descriptions $C = \{c_1, c_2, \ldots, c_k\}$, the model computes similarity scores using the cosine similarity between the image embedding and each class text embedding. The similarity score $s_j$ for class $j$ is defined as:

$$s_j = \frac{V(x) \cdot T(c_j)}{\|V(x)\| \, \|T(c_j)\|}, \quad \text{for } j = 1, \ldots, k. \tag{4}$$

The predicted class $\hat{y}$ is then obtained by selecting the class with the highest similarity score:

$$\hat{y} = \arg\max_j s_j. \tag{5}$$

The formulation in Equations equation 4–equation 5 enables zero-shot classification, as the model can generalize to previously unseen categories based solely on their textual descriptions. Nevertheless, to enhance performance on a specific downstream task with a fixed set of class labels $\mathcal{C}$, the model can be further adapted by optimizing a task-specific classifier using the cross-entropy loss:

$$\mathcal{L}_{\text{CE}}(M(x, \mathcal{C}), y), \tag{6}$$

where $M(x, \mathcal{C})$ denotes the similarity-based prediction over the class set $\mathcal{C}$, and $y$ is the ground-truth label. A common Ilharco et al. (2022); Tang et al. (2023); Ortiz-Jimenez et al. (2024); Yadav et al. (2023); Yang et al. (2024; 2023) fine-tuning strategy involves freezing the text encoder $T$ and updating only the vision encoder $V$, which has been empirically shown to yield the best performance Ilharco et al. (2022).

## A.3   LIMITATIONS

Our work focuses on CLIP-like vision models and image classification tasks; its applicability to other architectures (e.g., large language models) remains to be explored.

By introducing BVs we offer efficient, intuitive ways to capture with Task Arithmetic (TA) the complex interplay between task vectors and backdoors in MM. That is why we focus on TA and not any other novel MM method. The applicability of the obtained results to other methods remains to be explored.

Our IBVS method is rather a simple defense baseline than a final robust solution to backdoor problems in MM. We introduce it for further defense studies, as it requires almost no computation overhead to the defender, and its assumptions are minimalistic.

### A.4 LLM USAGE

Adhering to ICLR LLM usage policy we hereby declare that we used LLMs to polish the writing of the manuscript, style tables and for finding additional related work connected with backdoor attacks on MM.

## B EXPERIMENTAL DETAILS

### B.1 CODE TO REPRODUCE OUR FINDINGS

The code repository structure is adapted from Zhang et al. (2024a), to enable simple use of our method as an add-on to existing backdoor attacks on MM.

### B.2 BACKDOOR TRIGGER CONFIGURATION

Backdoor triggers are defined by a fixed set of parameters. These control both their visual appearance and how they interact with the model during training (see Table 5).

For our main experiments we use:

- `Patch Size`: 22 x 22 ( 1% pixels of the image), following Zhang et al. (2024a),
- `Location`: default bottom-right. We also test different patch locations for backdoor transfer experiments: bottom-left, upper-right, upper-left,
- `Trigger Type`: we use both types of triggers ($\bigstar, \oplus$) for different attacks, and show that proposed SBV merging improves ASR for both of them. We use $\oplus$ triggers for our proposed defense IBVS (white square trigger and wavelet trigger from Badnets Gu et al. (2017)),
- `Target class`: We average all our main results on five different target classes. Table 6 presents selected target classes from adversary tasks in our experiments. Details on used ImageNet100 classes are in the code repository added to the appendix,
- `Optimization seed`: We use 15 different optimization seeds for out experiments on SBV merging (some needed to calculate SBV, other to create 10 different models for single-task merging experiments),
- $\alpha$ We set this parameter to 5.0 following previous work Zhang et al. (2024a).

Table 5: Trigger configuration parameters used for backdoor attack design.

| Parameter | Description |
|---|---|
| Patch Size | Spatial dimensions of the trigger (e.g., $16 \times 16 - 0.5\%$, $22 \times 22 - 1\%$) |
| Location | Position of the patch within the input (e.g., bottom-right) |
| Trigger Type | Inherent ($\bigstar$) or injected $\oplus$ (like fixed white square $\square$) used in BadNets Gu et al. (2017) |
| Target Class | Output class enforced by the backdoor attack when triggered |
| Optimization Seed | Random seed used in training $M_{\text{backdoored}}$ or inherent trigger ($\bigstar$) optimization |
| $\alpha$ | Loss weight parameter for backdoor attack loss in finetuning $M_{\text{backdoored}}$ |

### B.3 BACKDOOR TARGET CLASS

We present output class enforced by the backdoor attack used in our experiments in Table 6. We average all the main results between results obtained for these five target classes.

Table 6: Selection of the target class for each task.

| Task | Target Class | | | | |
|---|---|---|---|---|---|
| | 1 | 2 | 3 | 4 | 5 |
| CIFAR100 | beaver | dolphin | otter | seal | whale |
| ImageNet100 | american coot | harvestman | macaw | bittern | electric ray |

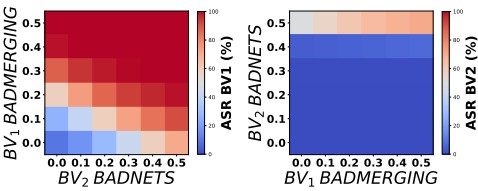

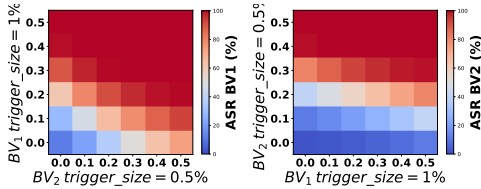

(a) Attack type

(b) Trigger size

Figure 9: **ViT-B-32: Backdoor transfer is asymmetric across attack types and trigger sizes:** $(\mathbf{BV_2} \to \mathbf{BV_1}) > (\mathbf{BV_1} \to \mathbf{BV_2})$. **(a)** BadNets $(BV_2)$ attack boosts the ASR of the BadMerging $(BV_1)$ attack, but not vice versa. **(b)** A smaller trigger attack (0.5% of image pixels) enhances the ASR of a $BV_1$ attack with a larger trigger (1%), while the reverse transfer is much weaker.

# C ADDITIONAL RESULTS

## C.1 ADDITIONAL RESULTS ON VIT-B-32

### C.1.1 ADDITIONAL RESULTS ON THE IMPORTANCE OF K-PARAMETER

Our findings (see Table 7) demonstrate that increasing the number of BVs used to form a single SBV leads to stronger and more robust attacks. Not only does the ASR improve with higher values of k (up to 5), but the results also become more stable - exhibiting reduced sensitivity to random noise during the merging process. This is reflected in the lower standard deviation reported across multiple runs.

Table 7: **Backdoor Merging: Effect of $k$ in Algorithm 2 under Single-Task with ViT-B.** Single-task, $\lambda = 0.1$, 10 tasks, TA. **Methods:** BM(★) — BadMerging

| Backdoor Attack | | | Adversary task: CIFAR100 | | | Adversary task: IMAGENET100 | | |
|---|---|---|---|---|---|---|---|---|
| Setting | Method | BV merging | CA | BA | ASR | CA | BA | ASR |
| | BM(★) | BV($k$=1) | 89.70 ± 0.03 | 89.74 ± 0.04 | 27.29 ± 12.14 | 85.88 ± 0.07 | **85.91** ± 0.04 | 84.01 ± 7.83 |
| | BM(★) | SBV($k$=2) | 89.70 ± 0.03 | **89.76 ± 0.03** | 55.09 ± 14.09 | 85.88 ± 0.07 | 85.84 ± 0.05 | 98.22 ± 0.89 |
| **Single-task** | BM(★) | SBV($k$=3) | 89.70 ± 0.03 | 89.73 ± 0.06 | 79.68 ± 10.98 | 85.88 ± 0.07 | 85.86 ± 0.07 | 99.71 ± 0.24 |
| | BM(★) | SBV($k$=4) | 89.70 ± 0.03 | 89.73 ± 0.06 | 91.95 ± 5.86 | 85.88 ± 0.07 | 85.87 ± 0.06 | 99.93 ± 0.05 |
| | BM(★) | SBV($k$=5) | 89.70 ± 0.03 | 89.69 ± 0.06 | **97.21 ± 2.30** | 85.88 ± 0.07 | 85.83 ± 0.09 | **99.99 ± 0.01** |

## C.1.2 IBVS PERFORMANCE

Despite its simplicity, IBVS demonstrates strong generalization across architectures such as ViT-B, ViT-L, and ConvNeXt. It consistently achieves substantial reductions in attack success rate (ASR), with performance degradation (in terms of backdoor accuracy, BA) typically below 1 percentage point.

Tables 8, 13 and 15 present results for ViT-B/ViT-L/ConvNext using varying numbers of BVs in SBV construction (parameter k). Note that BadMerging corresponds to BV(k=1). Reductions in ASR due to our defense are shown as $\Delta$ and are highlighted in bold.

## C.1.3 BACKDOOR TRANSFER

Figures 10 and 11 show the asymmetry of backdoor transfer given different trigger sizes across four different patch sizes (19 x 19, 22 x 22, 25 x 25, 28 x 28). Similarly to results from Figure 9b, smaller trigger patches enchance the ASR of bigger ones, and the reverse transfer is much weaker.

Table 8: **Backdoor Merging: Effect of IBVS Defense under Single-Task with ViT-B.** Single-task, $\lambda = 0.1$, 10 tasks, TA. BadMerging (BM, ★) with $\lambda_{IBVS} = 0.5$ when defense is enabled. k – number of BVs merged for an attack. **Methods:** BM(★) — BadMerging.

| Backdoor Attack | | | Adversary task: CIFAR100 | | Adversary task: IMAGENET100 | |
|---|---|---|---|---|---|---|
| Defense | Method | BV merging | BA | ASR | BA | ASR |
| - | BM(★) | BV($k$=1) | $89.74 \pm 0.04$ | $27.29 \pm 12.14$ | $85.91 \pm 0.04$ | $84.01 \pm 7.83$ |
| IBVS | BM(★) | BV($k$=1) | $89.06 \pm 0.11$ | $11.10 \pm 8.96$ | $85.20 \pm 0.12$ | $50.74 \pm 17.35$ |
| | | $\Delta$ (IBVS − none) | **-0.68** | **-16.19** | **-0.71** | **-33.27** |
| - | BM(★) | SBV($k$=2) | $89.76 \pm 0.03$ | $55.09 \pm 14.09$ | $85.84 \pm 0.05$ | $98.22 \pm 0.89$ |
| IBVS | BM(★) | SBV($k$=2) | $89.03 \pm 0.11$ | $26.17 \pm 15.53$ | $85.27 \pm 0.17$ | $84.45 \pm 10.19$ |
| | | $\Delta$ (IBVS − none) | **-0.73** | **-28.92** | **-0.57** | **-13.77** |
| - | BM(★) | SBV($k$=3) | $89.73 \pm 0.06$ | $79.68 \pm 10.98$ | $85.86 \pm 0.07$ | $99.71 \pm 0.24$ |
| IBVS | BM(★) | SBV($k$=3) | $88.98 \pm 0.07$ | $47.09 \pm 20.33$ | $85.34 \pm 0.21$ | $96.42 \pm 3.17$ |
| | | $\Delta$ (IBVS − none) | **-0.75** | **-32.59** | **-0.52** | **-3.29** |
| - | BM(★) | SBV($k$=4) | $89.73 \pm 0.06$ | $91.95 \pm 5.86$ | $85.87 \pm 0.06$ | $99.93 \pm 0.05$ |
| IBVS | BM(★) | SBV($k$=4) | $89.00 \pm 0.11$ | $64.91 \pm 20.54$ | $85.22 \pm 0.22$ | $99.09 \pm 0.95$ |
| | | $\Delta$ (IBVS − none) | **-0.73** | **-27.04** | **-0.65** | **-0.84** |
| - | BM(★) | SBV($k$=5) | $89.69 \pm 0.06$ | $97.21 \pm 2.30$ | $85.83 \pm 0.09$ | $99.99 \pm 0.01$ |
| IBVS | BM(★) | SBV($k$=5) | $89.00 \pm 0.11$ | $64.91 \pm 20.54$ | $85.22 \pm 0.22$ | $99.09 \pm 0.95$ |
| | | $\Delta$ (IBVS − none) | **-0.69** | **-32.30** | **-0.61** | **-0.90** |

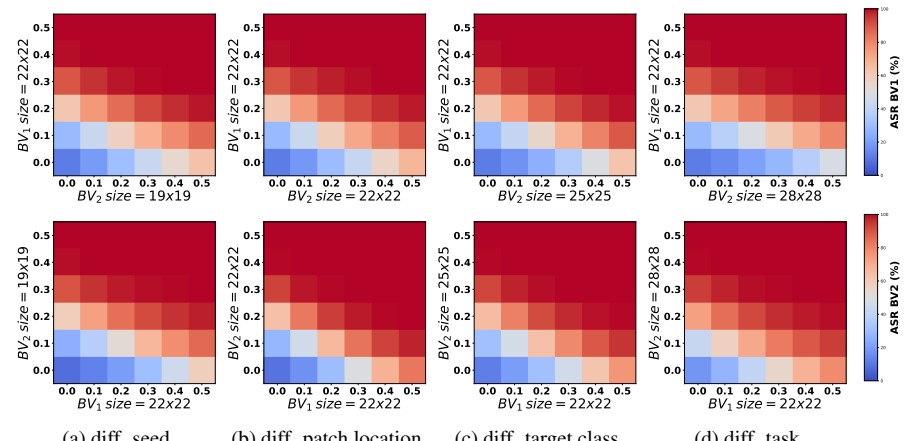

(a) diff. seed    (b) diff. patch location    (c) diff. target class    (d) diff. task

Figure 10: **CIFAR100 : Backdoor Transfer Asymmetry For Different Trigger Sizes.** Axes show $\lambda_{BV_1}, \lambda_{BV_2}$; A smaller trigger patch attack (e.g. 19x19 pixels) enhances the ASR of an attack with a larger trigger (e.g. 22x22 pixels), while the reverse transfer is much weaker.

### C.1.4 SBV: MULTI-TASK SETUP

Figure 12 shows how the adversary may use inherent triggers to attack the model even without contributing poisoned $M_{\text{backdoored}}$ during merging process. We show using ACC/ASR trajectories that inherent triggers are much more resilient in multi-task setup, than in single-task setup (see Figure 8). Tasks are different from each other and have cosine similarity close to zero, which results in small overlap in task vectors. Since different tasks interfere less with inherent trigger, the final attack success rate is high.

Table 9 show comparison between single- and multi-task scenarios for 6 tasks. Our SBV merging method improves the ASR in all tested scenarios. It is the first method that enables even the most simple, classical backdoor attack with injected triggers to withstand the merging process.

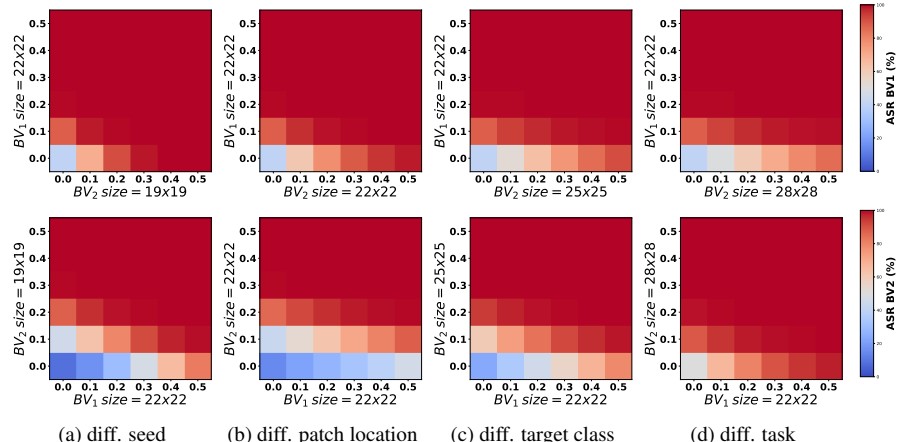

(a) diff. seed     (b) diff. patch location     (c) diff. target class     (d) diff. task

Figure 11: **ImageNet100: Backdoor Transfer Asymmetry For Different Trigger Sizes.** Axes show $\lambda_{BV_1}$, $\lambda_{BV_2}$; A smaller trigger patch attack (e.g. 19x19 pixels) enhances the ASR of an attack with a larger trigger (e.g. 22x22 pixels), while the reverse transfer is much weaker.

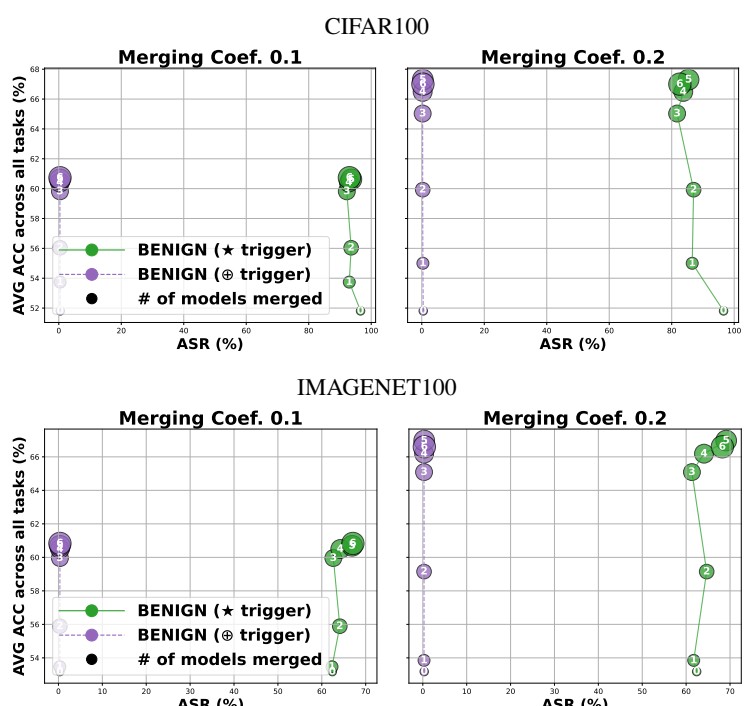

Figure 12: **ACC/ASR trajectories for inherent (⋆) and injected (⊕) triggers in all-clean multi-task MM.** Once again ⋆ triggers maintain their effectiveness even after merging multiple $M_{\text{clean}}$ models, while ⊕ triggers are completely ineffective in the absence of $M_{\text{backdoored}}$ model. Contrary to Figure 8, in multi-task setup inherent triggers do not weaken with more models merged, as tasks are much more independent and do not interfere with adversary task (cosine similarities between different TVs are close to 0.)

### C.1.5   IBVS: $\lambda$ PARAMETER

Table 10 shows the impact of $\lambda$ parameter for IBVS. Stronger defense merging coefficient ($\lambda_{IBVS}$) decrease the ASR more, but it comes with the cost of small final accuracy decrease. With smaller values of $\lambda_{IBVS}$, like 0.1 the defender can still lower the backdoor ASR, without any loss on the accuracy.

Table 9: **Backdoor Merging: Single- and Multi-Task Attack Results.** We merge $1 \times \theta_{\text{backdoored}} + 5 \times \theta_{\text{clean}}$ with $\lambda = 0.2$, reporting mean ± std. dev. **BV merging:** $AVG$ — weight averaging **Methods:** BN($\oplus$) — BadNets, BM($\star$) — BadMerging. TA. Visual encoder: ViT-B-32. Tasks for Multi-task setting: Adversary task, Cars Krause et al. (2013), SUN397 Xiao et al. (2010), EuroSAT Helber et al. (2019), GTSRB Stallkamp et al. (2011) and Pets Parkhi et al. (2012).

| Backdoor Attack | | | Adversary task: CIFAR100 | | | Adversary task: IMAGENET100 | | |
|---|---|---|---|---|---|---|---|---|
| Setting | Method | BV merging | CA | BA | ASR | CA | BA | ASR |
| Single-task | BN($\oplus$) | - | 89.43 ± 0.06 | 89.45 ± 0.03 | 0.25 ± 0.13 | 85.41 ± 0.11 | 85.54 ± 0.06 | 0.40 ± 0.16 |
| | BN($\oplus$) | AVG | 89.43 ± 0.06 | **89.51 ± 0.03** | 0.24 ± 0.13 | 85.41 ± 0.11 | **85.55 ± 0.08** | 0.41 ± 0.17 |
| | BN($\oplus$) | $SBV_{RDM}$ (Ours) | 89.43 ± 0.06 | 89.43 ± 0.04 | 0.48 ± 0.21 | 85.41 ± 0.11 | 85.43 ± 0.10 | 0.94 ± 0.24 |
| | BN($\oplus$) | $SBV_{SC}$ (Ours) | 89.43 ± 0.06 | 88.93 ± 0.07 | **100.00 ± 0.00** | 85.41 ± 0.11 | 84.86 ± 0.16 | **100.00 ± 0.00** |
| | BM($\star$) | - | 89.43 ± 0.06 | 89.49 ± 0.04 | 34.94 ± 15.00 | 85.41 ± 0.11 | **85.60 ± 0.08** | 96.95 ± 1.79 |
| | BM($\star$) | AVG | 89.43 ± 0.06 | **89.53 ± 0.05** | 35.55 ± 15.12 | 85.41 ± 0.11 | 85.52 ± 0.03 | 97.38 ± 1.54 |
| | BM($\star$) | $SBV_{RDM}$ (Ours) | 89.43 ± 0.06 | 89.49 ± 0.08 | 55.16 ± 15.88 | 85.41 ± 0.11 | 85.51 ± 0.11 | 99.34 ± 0.57 |
| | BM($\star$) | $SBV_{SC}$ (Ours) | 89.43 ± 0.06 | 89.20 ± 0.08 | **100.00 ± 0.00** | 85.41 ± 0.11 | 85.18 ± 0.08 | **100.00 ± 0.00** |
| Multi-task | BN($\oplus$) | - | 66.93 ± 0.11 | 66.88 ± 0.07 | 2.04 ± 1.59 | 66.60 ± 0.13 | 66.54 ± 0.07 | 0.67 ± 0.42 |
| | BN($\oplus$) | AVG | 66.93 ± 0.11 | **66.94 ± 0.05** | 2.07 ± 1.59 | 66.60 ± 0.13 | 66.56 ± 0.04 | 0.69 ± 0.41 |
| | BN($\oplus$) | $SBV_{RDM}$ (Ours) | 66.93 ± 0.11 | 66.88 ± 0.07 | 18.5 ± 10.96 | 66.60 ± 0.13 | 66.54 ± 0.07 | 2.59 ± 1.05 |
| | BN($\oplus$) | $SBV_{SC}$ (Ours) | 66.93 ± 0.11 | 65.66 ± 0.31 | **100.0 ± 0.0** | 66.60 ± 0.13 | 65.23 ± 0.2 | **100.0 ± 0.0** |
| | BM($\star$) | - | 66.93 ± 0.11 | 66.99 ± 0.06 | 99.92 ± 0.06 | 66.60 ± 0.13 | 66.67 ± 0.04 | 99.89 ± 0.15 |
| | BM($\star$) | AVG | 66.93 ± 0.11 | 67.09 ± 0.05 | 99.92 ± 0.07 | 66.60 ± 0.13 | **66.69 ± 0.05** | 99.92 ± 0.1 |
| | BM($\star$) | $SBV_{RDM}$ (Ours) | 66.93 ± 0.11 | **67.11 ± 0.05** | 99.99 ± 0.01 | 66.60 ± 0.13 | 66.66 ± 0.07 | 100.0 ± 0.01 |
| | BM($\star$) | $SBV_{SC}$ (Ours) | 66.93 ± 0.11 | 66.27 ± 0.14 | **100.0 ± 0.0** | 66.60 ± 0.13 | 65.91 ± 0.13 | **100.0 ± 0.0** |

Table 10: **Backdoor Defense: Injected BV Subtraction (IBVS).** Single-task results for IBVS variants across $\lambda_{IBVS} \in \{0.1, 0.3\}$, reporting mean BA and ASR. BV merging: $AVG$ — weight averaging. IBVS: $IBVS_\square$ uses a fixed white-square trigger; $IBVS_{BN}$ uses a wavelet trigger from BadNets. Backbone: ViT-B-32. No defense BV: Badmerging attack. We improve state-of-the-art ASR for BV creating stronger attack using our SBV merging method (columns). At the same time we propose a simple defense method (IBVS) to weaken the backdoors during merging and show the decrease of the ASR (rows). Values for $\lambda_{IBVS} == 0.3$ we report in the main part of the work.

| Method | $\lambda_{IBVS}$ | CIFAR100 | | | | | | | | ImageNet100 | | | | | | | |
|---|---|---|---|---|---|---|---|---|---|---|---|---|---|---|---|---|---|
| | | BV | | AVG | | $SBV_{RND}$ | | $SBV_{SC}$ | | BV | | AVG | | $SBV_{RND}$ | | $SBV_{SC}$ | |
| | | BA | ASR | BA | ASR | BA | ASR | BA | ASR | BA | ASR | BA | ASR | BA | ASR | BA | ASR |
| No defense | 0 | 89.74 | 27.29 | 89.72 | 27.56 | 89.72 | 35.93 | 89.69 | 97.21 | 85.91 | 84.01 | 85.96 | 84.79 | 85.97 | 92.55 | 85.83 | 99.99 |
| IBVS$_\square$ (Ours) | 0.1 | 89.82 | 25.60 | 89.87 | 26.78 | 89.89 | 34.81 | 89.69 | 96.75 | 85.93 | 78.84 | 85.65 | 84.00 | 85.69 | 91.92 | 85.77 | 99.96 |
| | 0.3 | 89.58 | 20.65 | 89.36 | 23.59 | 89.41 | 30.81 | 89.61 | 94.69 | 85.53 | 66.83 | 85.13 | 81.22 | 85.15 | **89.74** | 85.47 | **99.83** |
| IBVS$_{BN}$ (Ours) | 0.1 | 89.77 | 23.55 | 89.86 | 24.64 | 89.84 | 32.38 | 89.66 | 95.45 | 85.88 | 78.59 | 85.73 | 83.88 | 85.74 | 91.99 | 85.79 | 99.97 |
| | 0.3 | 89.57 | **16.56** | 89.41 | **19.36** | 89.42 | **25.37** | 89.46 | **89.21** | 85.61 | **65.26** | 85.14 | **80.89** | 85.14 | 90.17 | 85.59 | 99.89 |

## C.2 RESULTS ON ViT-L-14 AND CONVNEXT

Figures 13 and Table 12 show the results of our main experiments for larger visual encoder ViT-L-14 (we use mostly ViT-B-32 in the main part of our work). Our main results are consistent across these architectures. ViT-L-14 is more robust than ViT-B-32 to designed backdoor attacks. We suspect that the main reason behind it is the size of tested triggers, which are beyond single patch for small 14x14 ViT-L' patches. Training on small patch sizes can increase the robustness of the model to backdoor attacks (it is hard to optimize strong trigger pattern smaller than 14x14), but it comes with much higher cost during the ViT-L training.

Table 11: **Backdoor Merging: Single-Task Attack Results with TIES.** Additional results under single-task setting with $\lambda = 0.1$, 10 tasks, TIES merging. **Methods:** BN($\oplus$) — BadNets, BM($\star$) — BadMerging, SBV — ours.

| Backdoor Attack | | | Adversary task: CIFAR100 | | | Adversary task: IMAGENET100 | | |
|---|---|---|---|---|---|---|---|---|
| Setting | Method | BV merging | CA | BA | ASR | CA | BA | ASR |
| Single-task | BN($\oplus$) | - | 87.82 ± 0.08 | **87.76 ± 0.04** | 0.23 ± 0.09 | 85.10 ± 0.07 | **85.15 ± 0.05** | 0.29 ± 0.17 |
| | BN($\oplus$) | SBV (Ours) | 87.82 ± 0.08 | 87.68 ± 0.10 | **18.07 ± 11.94** | 85.10 ± 0.07 | 84.81 ± 0.15 | **20.0 ± 9.24** |
| | BM($\star$) | - | 87.82 ± 0.08 | 87.77 ± 0.03 | 58.20 ± 13.26 | 85.10 ± 0.07 | 85.14 ± 0.05 | 67.36 ± 18.28 |
| | BM($\star$) | SBV (Ours) | 87.82 ± 0.08 | **87.82 ± 0.05** | **99.93 ± 0.07** | 85.10 ± 0.07 | 85.06 ± 0.09 | **99.98 ± 0.02** |

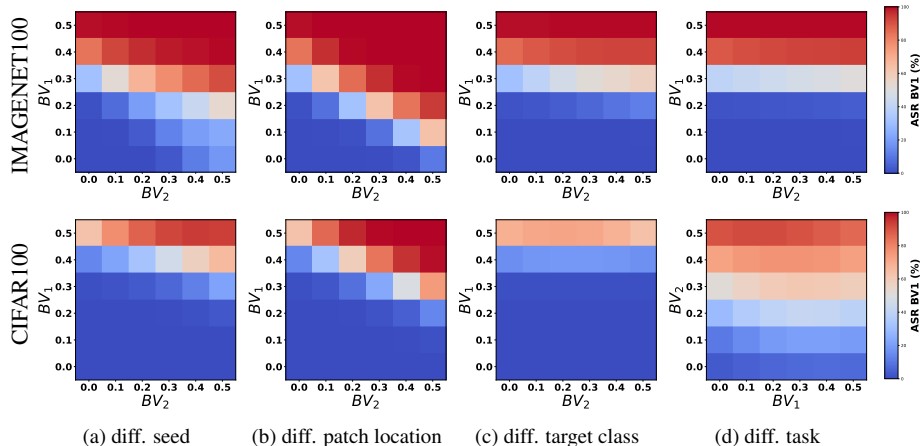

Figure 13: **Positive Backdoor Transfer Across Attacks** ($BV_2 \to BV_1$) **for ViT-L-14**. Adding $BV_2$ increases the ASR of $BV_1$. Axes show $\lambda_{BV_1}$, $\lambda_{BV_2}$; (a–c) use BVs from the same task (ImageNet100 or CIFAR100). The strongest transfer occurs across patch locations **(b)** and seeds **(a)**, but is weaker across target classes **(c)** and tasks **(d)**. In comparison to Figure 3 for ViT-L-14 different seeds has smaller backdoor transfer. We suspect this is connected with trigger size, as 22x22 trigger is bigger than ViT-L-14 training patches.

Table 12: **Backdoor Merging: Single-Task Attack Results with ViT-L-14 Visual Encoder.** We merge $1 \times \theta_{\text{backdoored}} + 9 \times \theta_{\text{clean}}$ with $\lambda = 0.1$, reporting mean ± std. dev. **BV merging:** $AVG$ — weight averaging **Methods:** BN($\oplus$) — BadNets, BM($\star$) — BadMerging.

| | Backdoor Attack | | Adversary task: CIFAR100 | | | Adversary task: IMAGENET100 | | |
|---|---|---|---|---|---|---|---|---|
| Setting | Method | BV merging | CA | BA | ASR | CA | BA | ASR |
| | BN($\oplus$) | - | 93.82 ± 0.03 | **93.86 ± 0.03** | 0.09 ± 0.04 | 91.20 ± 0.09 | **91.22 ± 0.03** | 0.11 ± 0.07 |
| | BN($\oplus$) | AVG | 93.82 ± 0.03 | 93.85 ± 0.03 | 0.09 ± 0.04 | 91.20 ± 0.09 | 91.21 ± 0.03 | 0.1 ± 0.06 |
| | BN($\oplus$) | $SBV_{RDM}$ (Ours) | 93.82 ± 0.03 | 93.7 ± 0.07 | 0.11 ± 0.03 | 91.20 ± 0.09 | 91.19 ± 0.06 | 0.63 ± 0.68 |
| | BN($\oplus$) | $SBV_{SC}$ (Ours) | 93.82 ± 0.03 | 93.38 ± 0.07 | **17.06 ± 16.89** | 91.20 ± 0.09 | 91.11 ± 0.13 | **81.86 ± 8.86** |
| Single-task | BM($\star$) | - | 93.82 ± 0.03 | 93.85 ± 0.03 | 0.1 ± 0.04 | 91.20 ± 0.09 | 91.19 ± 0.03 | 0.27 ± 0.17 |
| | BM($\star$) | AVG | 93.82 ± 0.03 | 93.82 ± 0.06 | 0.09 ± 0.04 | 91.20 ± 0.09 | **91.22 ± 0.02** | 0.26 ± 0.18 |
| | BM($\star$) | $SBV_{RDM}$ (Ours) | 93.82 ± 0.03 | 93.7 ± 0.08 | 0.16 ± 0.1 | 91.20 ± 0.09 | 91.2 ± 0.04 | 3.98 ± 3.73 |
| | BM($\star$) | $SBV_{SC}$ (Ours) | 93.82 ± 0.03 | 93.42 ± 0.03 | **32.21 ± 18.17** | 91.20 ± 0.09 | 91.01 ± 0.11 | **94.1 ± 5.08** |

## D  EVALUATION FOR ADDITIONAL BACKDOOR ATTACKS

Table 13: **Backdoor Merging: Effect of IBVS Defense under Single-Task with ViT-L.** Single-task, $\lambda = 0.1$, 10 tasks, TA. BadMerging (BM, $\star$) with $\lambda_{\text{IBVS}} = 0.5$ when defense is enabled. k – number of BVs merged for an attack. **Methods:** BM($\star$) — BadMerging.

| | Backdoor Attack | | Adversary task: CIFAR100 | | Adversary task: IMAGENET100 | |
|---|---|---|---|---|---|---|
| Defense | Method | BV merging | BA | ASR | BA | ASR |
| - | BM($\star$) | BV(k=1) | 93.85 ± 0.03 | 0.10 ± 0.04 | 91.19 ± 0.03 | 0.27 ± 0.17 |
| IBVS | BM($\star$) | BV(k=1) | 92.72 ± 0.02 | 0.12 ± 0.08 | 90.99 ± 0.09 | 0.21 ± 0.12 |
| | | $\Delta$ (IBVS − none) | **-1.13** | **+0.02** | **-0.20** | **-0.06** |
| - | BM($\star$) | SBV(k=5) | 93.42 ± 0.03 | 32.21 ± 18.17 | 91.01 ± 0.11 | 94.10 ± 5.08 |
| IBVS | BM($\star$) | SBV(k=5) | 92.47 ± 0.00 | 1.06 ± 0.93 | 90.89 ± 0.17 | 56.79 ± 18.42 |
| | | $\Delta$ (IBVS − none) | **-0.95** | **-31.15** | **-0.12** | **-37.31** |

Table 14: **Backdoor Merging: Single-Task Attack Results with ConvNeXt Backbone.** We merge $1 \times \theta_{\text{backdoored}} + 9 \times \theta_{\text{clean}}$ with $\lambda = 0.1$, reporting mean ± std. dev. **BV merging:** -— no merging. **Methods:** BN($\oplus$) — BadNets, BM($\star$) — BadMerging, SBV — ours.

| Setting | Backdoor Attack | | Adversary task: CIFAR100 | | | Adversary task: IMAGENET100 | | |
|---|---|---|---|---|---|---|---|---|
| | Method | BV merging | CA | BA | ASR | CA | BA | ASR |
| Single-task | BN($\oplus$) | - | 90.62 ± 0.03 | 90.71 ± 0.02 | 1.06 ± 1.74 | 88.63 ± 0.05 | **88.73 ± 0.04** | 0.16 ± 0.07 |
| | BN($\oplus$) | SBV (Ours) | 90.62 ± 0.03 | 90.71 ± 0.04 | **100.0 ± 0.01** | 88.63 ± 0.05 | 88.61 ± 0.02 | **99.52 ± 0.58** |
| | BM($\star$) | - | 90.62 ± 0.03 | 90.71 ± 0.05 | 11.03 ± 3.71 | 88.63 ± 0.05 | 88.71 ± 0.02 | 3.97 ± 1.57 |
| | BM($\star$) | SBV (Ours) | 90.62 ± 0.03 | **90.76 ± 0.02** | **100.0 ± 0.00** | 88.63 ± 0.05 | 88.53 ± 0.09 | **99.99 ± 0.01** |

Table 15: **Backdoor Merging: Effect of IBVS Defense under Single-Task with ConvNeXt.** Single-task, $\lambda = 0.1$, 10 tasks, TA. BadMerging (BM, $\star$) with $\lambda_{\text{IBVS}} = 0.5$ when defense is enabled. k – number of BVs merged for an attack. **Methods:** BM($\star$) — BadMerging.

| Backdoor Attack | | | Adversary task: CIFAR100 | | Adversary task: IMAGENET100 | |
|---|---|---|---|---|---|---|
| Defense | Method | BV merging | BA | ASR | BA | ASR |
| - | BM($\star$) | BV($k=1$) | 90.62 ± 0.03 | 11.03 ± 3.71 | 88.63 ± 0.05 | 3.97 ± 1.57 |
| IBVS | BM($\star$) | BV($k=1$) | 90.06 ± 0.03 | 6.91 ± 4.60 | 88.39 ± 0.02 | 1.39 ± 0.83 |
| | | $\Delta$ (IBVS − none) | **-0.56** | **-4.12** | **-0.24** | **-2.58** |
| - | BM($\star$) | SBV($k=5$) | 90.62 ± 0.03 | 100.0 ± 0.00 | 88.63 ± 0.05 | 99.99 ± 0.01 |
| IBVS | BM($\star$) | SBV($k=5$) | 90.13 ± 0.12 | 96.99 ± 3.62 | 88.37 ± 0.08 | 72.98 ± 23.77 |
| | | $\Delta$ (IBVS − none) | **-0.49** | **-3.01** | **-0.26** | **-27.01** |

To validate the generalization capability of our approach beyond standard patch-based triggers, we extended our evaluation to include three distinct backdoor paradigms:

- **WaNet:** A warping-based backdoor that applies subtle, smooth geometric distortions across the entire image field, avoiding localized patch triggers.
- **Blend:** A global blending attack where the trigger pattern is softly interpolated with the entire input image, rendering the trigger invisible to patch detection.
- **BadTV:** A task-vector-based composite backdoor generated through arithmetic operations (e.g., addition and subtraction of poisoned task vectors) to engineer a robust backdoor for addition and negation operations.

**Experimental Setup:** The experimental configuration remains consistent with the baseline setup reported in previous sections (ViT-B backbone, merging 10 tasks, merging coefficient $\lambda = 0.1$).

### D.1 SBV AND IBVS PERFORMANCE ANALYSIS

Results presented in Table 16 demonstrates strong scalability of SBV across these diverse attack types. Crucially, the process of merging multiple sparse backdoor vectors consistently amplifies the potency of the attack. This cumulative enhancement is invariant to the underlying backdoor mechanism - whether geometric (WaNet), additive (Blend), or arithmetic (BadTV) - confirming that the benefits of sparsity and vector merging are fundamental rather than specific to patch-based attacks.

IBVS strongly reduces SBV-WaNet and SBV-BadTV attacks, while it provides only limited reduction for SBV-Blend (see Table 17). For IBVS we use BadNets wavelet triggers as in the 3rd row in Table 4.

Table 16: Extension of Table 3: additional backdoor attacks with injected triggers: Blend, Wanet and BadTV.**ViT-B-32: Backdoor Merging: Single-Task Attack Results.** We merge $1 \times \theta_{\text{backdoored}} + 9 \times \theta_{\text{clean}}$ with $\lambda = 0.1$, reporting mean ± std. dev. **BV merging:** $AVG$ — weight averaging **Methods:** BN($\oplus$) — BadNets, BM($\star$) — BadMerging. ViT-B-32.

| Backdoor Attack | | | Adversary task: CIFAR100 | | | Adversary task: IMAGENET100 | | |
|---|---|---|---|---|---|---|---|---|
| Setting | Method | BV merging | CA | BA | ASR | CA | BA | ASR |
| | BN($\oplus$) | - | 89.70 ± 0.03 | 89.73 ± 0.03 | 0.2 ± 0.12 | 85.88 ± 0.07 | 85.88 ± 0.04 | 0.26 ± 0.1 |
| | BN($\oplus$) | AVG | 89.70 ± 0.03 | **89.75 ± 0.03** | 0.21 ± 0.12 | 85.88 ± 0.07 | **86.0 ± 0.03** | 0.27 ± 0.1 |
| | BN($\oplus$) | $SBV_{RDM}$ (Ours) | 89.70 ± 0.03 | 89.72 ± 0.04 | 0.23 ± 0.12 | 85.88 ± 0.07 | **86.0 ± 0.06** | 0.33 ± 0.14 |
| | BN($\oplus$) | $SBV_{SC}$ (Ours) | 89.70 ± 0.03 | 89.73 ± 0.12 | **12.23 ± 10.52** | 85.88 ± 0.07 | 85.8 ± 0.06 | **29.45 ± 12.36** |
| | Blend($\oplus$) | - | 89.70 ± 0.03 | **90.05 ± 0.06** | 1.81 ± 2.64 | 85.88 ± 0.07 | 86.26 ± 0.12 | 0.44 ± 0.56 |
| | Blend($\oplus$) | AVG | 89.70 ± 0.03 | 89.78 ± 0.05 | 1.06 ± 1.42 | 85.88 ± 0.07 | 85.98 ± 0.06 | 0.39 ± 0.42 |
| | Blend($\oplus$) | $SBV_{RDM}(Ours)$ | 89.70 ± 0.03 | 89.92 ± 0.17 | 48.92 ± 10.74 | 85.88 ± 0.07 | **86.27 ± 0.10** | 2.57 ± 2.38 |
| | Blend($\oplus$) | $SBV_{SC}(Ours)$ | 89.70 ± 0.03 | 89.53 ± 0.09 | **97.54 ± 0.52** | 85.88 ± 0.07 | 86.14 ± 0.08 | **55.43 ± 4.40** |
| | Wanet($\oplus$) | - | 89.70 ± 0.03 | **89.90 ± 0.05** | 0.17 ± 0.12 | 85.88 ± 0.07 | **86.22 ± 0.06** | 0.31 ± 0.30 |
| Single-task | Wanet($\oplus$) | AVG | 89.70 ± 0.03 | 89.77 ± 0.06 | 0.17 ± 0.10 | 85.88 ± 0.07 | 85.92 ± 0.14 | 0.30 ± 0.28 |
| | Wanet($\oplus$) | $SBV_{RDM}(Ours)$ | 89.70 ± 0.03 | **89.90 ± 0.05** | 0.21 ± 0.12 | 85.88 ± 0.07 | 86.18 ± 0.07 | 0.66 ± 0.56 |
| | Wanet($\oplus$) | $SBV_{SC}(Ours)$ | 89.70 ± 0.03 | 89.54 ± 0.11 | **39.32 ± 6.29** | 85.88 ± 0.07 | 85.94 ± 0.25 | **66.15 ± 3.98** |
| | BadTV($\oplus$) | - | 89.70 ± 0.03 | 89.68 ± 0.04 | 0.15 ± 0.11 | 85.88 ± 0.07 | **85.82 ± 0.05** | 0.20 ± 0.13 |
| | BadTV($\oplus$) | AVG | 89.70 ± 0.03 | **89.70 ± 0.02** | 0.14 ± 0.11 | 85.88 ± 0.07 | 85.71 ± 0.10 | 0.18 ± 0.12 |
| | BadTV($\oplus$) | $SBV_{RDM}(Ours)$ | 89.70 ± 0.03 | 89.68 ± 0.02 | 0.15 ± 0.12 | 85.88 ± 0.07 | 85.76 ± 0.04 | 0.20 ± 0.14 |
| | BadTV($\oplus$) | $SBV_{SC}(Ours)$ | 89.70 ± 0.03 | 89.18 ± 0.11 | **0.16 ± 0.12** | 85.88 ± 0.07 | 85.17 ± 0.09 | **0.21 ± 0.14** |
| | BM($\star$) | - | 89.70 ± 0.03 | **89.74 ± 0.04** | 27.30 ± 12.15 | 85.88 ± 0.07 | 85.91 ± 0.04 | 84.00 ± 7.83 |
| | BM($\star$) | AVG | 89.70 ± 0.03 | 89.50 ± 0.04 | 13.84 ± 9.70 | 85.88 ± 0.07 | 85.53 ± 0.04 | 78.00 ± 9.16 |
| | BM($\star$) | $SBV_{RDM}(Ours)$ | 89.70 ± 0.03 | 89.72 ± 0.02 | 36.27 ± 13.53 | 85.88 ± 0.07 | **85.93 ± 0.07** | 92.59 ± 3.74 |
| | BM($\star$) | $SBV_{SC}(Ours)$ | 89.70 ± 0.03 | 89.69 ± 0.06 | **97.21 ± 2.30** | 85.88 ± 0.07 | 85.83 ± 0.09 | **99.99 ± 0.01** |

Table 17: **IBVS Defense Evaluation on Additional Backdoor Attacks.** We report the impact of the defense scaling coefficient $\lambda_1$ on Blend, WaNet, and BadTV attacks. Rows labeled $\Delta$ *vs 0* show the absolute change in BA and ASR compared to the no-defense baseline ($\lambda_1 = 0$). Bold values in $\Delta$ rows indicate the reduction in ASR.

| Attack | Setting ($\lambda_1$) | Adversary task: CIFAR100 | | Adversary task: ImageNet100 | |
|---|---|---|---|---|---|
| | | BA | ASR | BA | ASR |
| Blend | 0.1 | 89.52 | 97.44 | 86.10 | 54.60 |
| | $\Delta$ *vs 0* | -0.02 | **-0.10** | -0.04 | **-0.82** |
| | 0.3 | 89.40 | 97.14 | 86.02 | 52.77 |
| | $\Delta$ *vs 0* | -0.14 | **-0.40** | -0.12 | **-2.65** |
| WaNet | 0.1 | 89.52 | 38.23 | 85.94 | 61.90 |
| | $\Delta$ *vs 0* | -0.02 | **-1.09** | 0.00 | **-4.25** |
| | 0.3 | 89.32 | 33.91 | 85.85 | 44.27 |
| | $\Delta$ *vs 0* | -0.22 | **-5.41** | -0.09 | **-21.88** |
| BadTV | 0.1 | 88.63 | 8.66 | 83.90 | 11.05 |
| | $\Delta$ *vs 0* | -0.03 | **-2.56** | -0.04 | **-1.06** |
| | 0.3 | 88.49 | 5.29 | 83.78 | 9.15 |
| | $\Delta$ *vs 0* | -0.17 | **-5.93** | -0.16 | **-2.96** |

# E MAGNITUDE-BASED BACKDOOR VECTOR PRUNING ANALYSIS

We show that BV framework generalize over backdoors created by training data poisoning (less diffused backdoor signal) and training time poisoning (more diffused) as well as over injected (less diffused) and inherent ones (more diffused).

To investigate the distribution of backdoor information within the task vectors, we employ a magnitude-based pruning procedure. In this setting, the parameter $x\%$ denotes the retention ratio, where we explicitly keep the $x\%$ of weights possessing the *smallest magnitudes* and prune the remaining $(100 - x)\%$. Consequently, increasing $x\%$ corresponds to the gradual reintroduction of weights with increasingly larger magnitudes (i.e., those that deviate most significantly from the pre-trained model). This controlled ablation allows us to determine whether the backdoor behavior is primarily driven by large-magnitude parameters.

Table 18: **ASR under Magnitude Pruning.** We report ASR as a function of $x\%$, where $x$ represents the fraction of the *smallest-magnitude* weights retained.

| Method | Setting | Percentage of Smallest Weights Retained ($x\%$) | | | | | | | | | |
|---|---|---|---|---|---|---|---|---|---|---|---|
| | | 0 | 50 | 55 | 60 | 65 | 70 | 75 | 80 | 85 | 90 |
| **BadNets** | Data Poisoning | - | 0.23 | 0.27 | 0.32 | 0.39 | 0.54 | 0.75 | 1.37 | 4.87 | 57.19 |
| | Train Poisoning | 0.25 | 0.19 | 0.19 | 0.21 | 0.22 | 0.26 | 0.41 | 2.46 | 42.40 | 99.82 |
| **BadMerging** | BV | 10.02 | 15.25 | 20.36 | 28.22 | 41.09 | 58.97 | 79.72 | 94.96 | 99.76 | 100.00 |
| | SBV (Ours) | 10.02 | 10.02 | 10.02 | 10.02 | 10.02 | 10.02 | 22.40 | 87.71 | 100.00 | 100.00 |

Most pruning works show results for data poisoning with backdoor samples (as a most general threat). In model merging there exists an additional threat of adding a backdoor differently: in this work, following BadMerging we add backdoors during training procedure. As a consequence, the backdoor signal is scattered more broadly than in data-poisoning backdoor approaches.

Table 18 presents the ASR across different retention thresholds. We observe distinct behaviors between attack methods: BadNets (BN) triggers remain dormant at lower retention rates, spiking only when $x \geq 85\%$, suggesting reliance on high-magnitude weights. In contrast, BadMerging (BM) exhibits a steady rise in ASR much earlier ($x \approx 60\%$), indicating a more diffuse backdoor injection.

In SBV backdoor information is strengthened in a smaller number of weights, and non-backdoor information is explicitly removed with (sign-consistency) heuristic. Although future heuristics may improve this, we show that even this simple approach yields a much more merging-resilient attack by explicitly removing less relevant BV components.

By BV pruning we show that:

1. Backdoor signal is strong in BV, even after pruning 10% of the highest-magnitude weights,

2. Backdoor Signal in BV can be distributed and strong even in smaller-magnitude weights (for inherent backdoor attacks),

3. The distribution of backdoor signal in small weights is stronger when backdoor is implanted during training compared to backdooring via data poisoning (see Table 18).

Our BV framework generalizes across different backdoor implanting settings while preserving their distinguishing characteristics.

## F  ADAPTIVE ATTACK EVALUATION

To rigorously assess the robustness of our defense, we evaluate an adaptive attack scenario where the adversary attempts to counteract the Injected BV Subtraction (IBVS).

### F.1  ADVERSARY MODEL AND ASSUMPTIONS

We operate under the following assumptions regarding the adversary's knowledge and capabilities:

- **(A1) Knowledge:** The adversary possesses full knowledge of the specific trigger pattern and the proxy dataset used by the defender for IBVS.
- **(A2) Capability:** The adversary controls only a single model within the merging pool (consistent with our single-task setting).
- **(A3) Defender's Parameters:** The defender selects the merging coefficient $\lambda$ and the defense scaling coefficient $\lambda_1$.
- **(A4) Adversary's Parameters:** The adversary selects a scaling coefficient $\lambda_2$ to inject an adaptive counter-vector.

**Attack Mechanism:** The adversary computes an adaptive task vector, denoted as $\mathbf{BV}'$, using the known trigger and dataset. To neutralize the anticipated subtraction of the defense vector by the

defender, the adversary adds $\lambda_2 \cdot \mathbf{BV}'$ to their backdoored model weights. This effectively results in a multi-trigger attack, implanting both the original backdoor (e.g., BadMerging) and the adaptive backdoor (e.g., BadNets).

## F.2 EMPIRICAL ANALYSIS

We evaluate the interaction between the Adaptive Attack (AA) and IBVS defense by varying the scaling coefficients. Our findings, summarized in Table 19, reveal the following insights:

1. **Baseline Impact:** In the absence of defense ($\lambda_1 = 0$), deploying the adaptive vector (AA) slightly increases the ASR of the primary attack.

2. **Defense Mitigation:** When IBVS is active ($\lambda_1 > 0$), the adaptive attack ($\lambda_2 > 0$) can partially reduce the defense's effectiveness compared to the non-adaptive setting. However, it fails to fully negate the defense.

3. **Adversarial Limitations:** Crucially, the adversary cannot overpower the defense. The defender's cumulative effect scales with the number of models, whereas the adversary's influence is bounded by the single injected model. Specifically, the defense holds as long as $\lambda_2 \cdot \lambda < \lambda_1 \cdot \lambda \cdot N$, where $N$ is the number of models.

Table 19 details the specific performance shifts. While the adaptive attack marginally recovers ASR against standard BV merging, the **SBV** method remains highly robust against IBVS, maintaining near-perfect attack success rates.

Table 19: **Adaptive Attack (AA) vs. IBVS Defense.** We report BA and ASR for varying merging coefficients for defense vector ($\lambda_1$) and adversarial adaptations ($\lambda_2$). **AA**: Adaptive Attack present ($\lambda_2 > 0$). **IBVS**: Defense present ($\lambda_1 > 0$). The task merging coefficient is fixed at $\lambda = 0.1$.

| Setting (AA / IBVS) | Params ($\lambda_1 / \lambda_2$) | CIFAR100 BA / ASR | $\Delta$BA | $\Delta$ASR | ImageNet100 BA / ASR | $\Delta$BA | $\Delta$ASR |
|---|---|---|---|---|---|---|---|
| \multicolumn{8}{c}{**BadMerging (BV)**} | | | | | | | |
| ✗ / ✗ | 0.0 / 0.0 | 89.74 / 27.29 | 0.00 | 0.00 | 85.91 / 84.01 | 0.00 | 0.00 |
| ✓ / ✗ | 0.0 / 0.5 | 89.71 / 27.89 | -0.03 | +0.60 | 85.88 / 85.98 | -0.03 | +1.97 |
| ✓ / ✗ | 0.0 / 1.0 | 89.70 / 28.44 | -0.04 | +1.15 | 85.94 / 87.71 | +0.03 | +3.70 |
| ✗ / ✓ | 0.1 / 0.0 | 89.82 / 25.60 | +0.08 | -1.69 | 85.93 / 78.84 | +0.02 | -5.17 |
| ✓ / ✓ | 0.1 / 0.5 | 89.78 / 26.98 | +0.04 | -0.31 | 85.93 / 82.81 | +0.02 | -1.20 |
| ✓ / ✓ | 0.1 / 1.0 | 89.71 / 27.63 | -0.03 | +0.34 | 85.88 / 84.88 | -0.03 | +0.87 |
| ✗ / ✓ | 0.3 / 0.0 | 89.58 / 20.65 | -0.16 | -6.64 | 85.53 / 66.83 | -0.38 | -17.18 |
| ✓ / ✓ | 0.3 / 0.5 | 89.70 / 24.16 | -0.04 | -3.13 | 85.61 / 74.62 | -0.30 | -9.39 |
| ✓ / ✓ | 0.3 / 1.0 | 89.70 / 25.14 | -0.04 | -2.15 | 85.66 / 77.25 | -0.25 | -6.76 |
| \multicolumn{8}{c}{**BadMerging (SBV)**} | | | | | | | |
| ✗ / ✗ | 0.0 / 0.0 | 89.69 / 97.21 | 0.00 | 0.00 | 85.83 / 99.99 | 0.00 | 0.00 |
| ✓ / ✗ | 0.0 / 0.5 | 89.65 / 97.32 | -0.04 | +0.11 | 85.85 / 100.00 | +0.02 | +0.01 |
| ✓ / ✗ | 0.0 / 1.0 | 89.68 / 97.37 | -0.01 | +0.16 | 85.84 / 100.00 | +0.01 | +0.01 |
| ✗ / ✓ | 0.1 / 0.0 | 89.69 / 96.75 | 0.00 | -0.46 | 85.77 / 99.96 | -0.06 | -0.03 |
| ✓ / ✓ | 0.1 / 0.5 | 89.74 / 96.85 | +0.05 | -0.36 | 85.84 / 99.98 | +0.01 | -0.01 |
| ✓ / ✓ | 0.1 / 1.0 | 89.72 / 97.00 | +0.03 | -0.21 | 85.80 / 99.99 | -0.03 | 0.00 |
| ✗ / ✓ | 0.3 / 0.0 | 89.61 / 94.69 | -0.08 | -2.52 | 85.47 / 99.83 | -0.36 | -0.16 |
| ✓ / ✓ | 0.3 / 0.5 | 89.63 / 94.74 | -0.06 | -2.47 | 85.53 / 99.94 | -0.30 | -0.05 |
| ✓ / ✓ | 0.3 / 1.0 | 89.61 / 95.19 | -0.08 | -2.02 | 85.57 / 99.96 | -0.26 | -0.03 |

Table 20: Extension of Table 4: comparison with adapted Federated Learning defense methods. **ViT-B-32 – Backdoor Defense: Injected BV Subtraction (IBVS).** Single-task defense results for merging $1 \times \theta_{\text{backdoored}} + 9 \times \theta_{\text{clean}}$ with $\lambda = 0.1$, reporting mean BA and ASR. Federated Learning adapted baselines: FLAME, KRUM. Backbone: ViT-B-32. No defense: BadMerging attack.

| Setting | Adversary task: CIFAR100 | | | | | | | | Adversary task: ImageNet100 | | | | | | | |
|---|---|---|---|---|---|---|---|---|---|---|---|---|---|---|---|---|
| | BV | | AVG | | $SBV_{RND}$ | | $SBV_{SC}$ | | BV | | AVG | | $SBV_{RND}$ | | $SBV_{SC}$ | |
| | BA | ASR | BA | ASR | BA | ASR | BA | ASR | BA | ASR | BA | ASR | BA | ASR | BA | ASR |
| No defense | 89.74 | 27.30 | 89.50 | 13.84 | 89.72 | 36.27 | 89.69 | 97.21 | **85.91** | 84.00 | 85.53 | 78.00 | **85.93** | 92.59 | 85.83 | 99.99 |
| IBVS□ (Ours) | 89.58 | 20.65 | 89.36 | 23.59 | 89.41 | 30.81 | 89.61 | 94.69 | 85.53 | 66.83 | 85.13 | 81.22 | 85.15 | 89.74 | 85.47 | 99.83 |
| IBVS$_{BN}$ (Ours) | 89.57 | 16.56 | 89.41 | 19.36 | 89.42 | 25.37 | 89.46 | 89.21 | 85.61 | 65.26 | 85.14 | 80.89 | 85.14 | 90.17 | 85.59 | 99.89 |
| FLAME | **89.75** | **10.21** | **89.76** | **10.22** | **89.76** | **10.22** | **89.76** | **10.22** | 85.87 | 39.63 | **85.86** | 39.63 | 85.86 | 39.63 | **85.86** | 39.63 |
| KRUM | 89.14 | 11.77 | 89.14 | 11.77 | 89.14 | 11.77 | 89.14 | 11.77 | 85.04 | **28.03** | 85.04 | **28.03** | 85.04 | **28.03** | 85.04 | **28.03** |

# G  COMPARISON TO FEDERATED LEARNING BASELINES

FL relies on model merging as its core aggregation mechanism, where locally trained updates from decentralized clients are fused to create a shared global model without accessing private data.

This relationship results in a similar vulnerability to backdoor attacks. The seminal work Bagdasaryan et al. (2018) demonstrates that even a small fraction of malicious participants can reliably inject backdoors into the global model. A number of follow-up works propose backdoor defense mechanisms for FL. KRUM Blanchard et al. (2017) selects the client model update with the minimum distance to the majority of other updates received in a training round, aiming to exclude malicious outliers. FLAME Nguyen et al. (2022) combines update filtering, adaptive clipping, and noising to mitigate the influence of backdoored updates. Wu et al. (2020) observes that backdoored models often lie in sharp, isolated minima that are not smoothly connected to the benign training trajectory, enabling their detection and filtering.

Despite the conceptual overlap between FL and MM, few MM backdoor works have so far adapted or evaluated established FL defenses. As a result, the substantial insights, threat models, and mitigation strategies developed within FL have not been systematically leveraged in the MM setting. While the fields are closely related, sometimes significant work remains in translating these defenses to the unique characteristics and constraints of model merging.

We present the comparison of IBVS to two adapted FL defense methods as additional baselines in Table 20. Our evaluation yields several key observations regarding the effectiveness of FLAME and KRUM compared to our proposed IBVS method:

- **Performance of Baselines:** Surprisingly, FLAME proved highly effective, achieving strong utility (High BA) while surpassing IBVS in mitigating Attack Success Rate (ASR). KRUM also successfully reduced ASR, though at a significant cost to the final model's accuracy (BA).

- **Mechanism of Action:** The fundamental difference lies in how these methods handle poisoned updates. The Federated Learning (FL) baselines use *filtering*: they minimize the impact of the Backdoor Vector (BV) by down-weighting or excluding suspicious model coefficients. In contrast, IBVS does not filter models but rather mitigates the backdoor's impact directly by exploiting the structural similarity of BVs to subtract the trigger signal.

- **Stability and Assumptions:** Both FL baselines exhibit higher variance (standard deviation) across results, likely due to their filtering mechanisms being sensitive to the specific target class distributions. Furthermore, they rely on stricter assumptions than IBVS: KRUM requires prior knowledge of the exact number of poisoned workers and struggles in multi-task settings, while FLAME's clustering assumption fails if more than $50\%$ of the tasks are poisoned.

- **Complementarity:** Finally, we note that these approaches are not mutually exclusive. Since FL baselines (that use filtering) and IBVS (subtraction) operate on different principles, they could theoretically be combined to achieve a compounded defensive effect.

