# OpenReview forum: "Backdoor Vectors: a Task Arithmetic View on Backdoor Attacks and Defenses in Model Merging"
_ICLR.cc/2026/Conference — ICLR 2026 Conference Desk Rejected Submission_

### Official Review · Reviewer_zgj3 · 2025-10-29

**Soundness:** 2
**Presentation:** 2
**Contribution:** 1
**Rating:** 2
**Confidence:** 4

**Summary:**

This paper proposes viewing backdoor attacks in CLIP-based image classification as task vectors (from task arithmetic work). The paper shows that adding backdoor vectors results in more potent backdoor attacks in the context of model merging with two selected backdoor attacks: badnets and bad merging. The paper also proposes that subtracting backdoor vectors is a good defense against unknown triggers.

**Strengths:**

- The paper presents some insights into backdoors in CLIP-based image classification tasks.
- The paper shows that a sign-consistent merging of backdoor vectors results in better backdoor attacks (even better than no merging scenarios from the results).
- The paper shows backdoor vectors follow task arithmetic and proposes to use subtraction as a defense.

**Weaknesses:**

- The novelty of this paper may be limited. The use of task vectors in the backdoor domain is not new, as cited in the paper (BADTV).
- The framing of backdoor vector merging is not practical. To get one backdoor vector, one needs to train one backdoor model first. It is not clear why there is a need for the attacker to train multiple backdoor models instead of training one potent backdoor model.
- The experiments are limited, and the effectiveness of the IBVS is not clear.
- The experiments are way less compared to the original task, arithmetic (TV), even for the image classification tasks.
There are only two backdoor attacks in the experiment. The paper may overclaim the generalization ability of IBVS.

Minor:
- The writing is not very coherent. The reviewer thinks the paper should focus on defense against backdoors, as it seems the best contribution of the paper.
- The shorthands, SC, RND are not clearly defined in the paper for Algorithm 1, although the reviewer understands them as sign consistent and random.
- Algorithms 1 and 2 are not explained in the paper.

**Questions:**

- How does this paper compare to BADTV, as both of them are inspired by task vectors?

- What is the target class (alone) accuracy on clean images after subtracting backdoor vectors?

- In Table 3, does it mean merging is better than not merging?  Is the bad merging ASR lower than before merging (27%)?

- In Table 4, are the results for the same badnets with different triggers?

- The paper says addition and subtraction of BVs are not symmetric. Then what are the guarantees of the backdoor defense (IBVS)?

- In Table 9, the adversary tasks from the caption are different from the ones inside the table. How do the multi-tasks work for the results?

---

> ### Author Response · Authors · 2025-11-25
>
> We thank the Reviewer for the valuable feedback. Below, we address the key questions and concerns: first, we provide a brief overview of the most important points, followed by detailed responses in the additional comments.
>
> **The three main concerns raised are:**
> - **W1)** whether our novelty is limited by the BadTV work,
> - **W2)** whether BV merging is practical, and
> - **W3)** whether the experiments - especially those for IBVS - are sufficiently extensive.
>
> ### Ad. W1: Novelty and comparison of our work to BadTV
> Both works examine backdoors in task-vector–based model adaptation. **BadTV (arXiv, 2025)** introduces an attack where two backdoored models are combined to produce a composite backdoor that remains effective even after model merging operations like addition or subtraction.
>
> On the other hand, Backdoor Vectors (ours) **frame the backdoor behaviour as a task itself**, and build **a framework to understand, analyze, and measure backdoor impact in model merging**. We further show the application of our framework including SBV attacks, IBVS defense and comparing different backdoors beyond standard ASR metric.
>
> Thus, while BadTV is a novel and interesting attack-focused work, **we provide a higher-level BV framework with broader applications**. Our work goes beyond prior approaches in several important ways:
>
> - We are the first to propose **a unified framework for viewing backdoor attacks and defenses through task vectors**, and to **demonstrate its practical usefulness** (via SBV, IBVS, and BV analysis).
> - We are the first to introduce **backdoor merging into a single, stronger attack (SBV)**, supported by extensive empirical evidence.
> - We are the first to show the **importance of inherent backdoors in model merging**, revealing e.g. that they are far more resilient than injected ones; we leverage BV insights in the IBVS defense.
> - We are the first to provide a **vector-space analysis of backdoors in model merging**, enabling measurements of backdoor interference beyond ASR. **The BV framework allows us to quantify these complex phenomena with concrete task-vector tools.**
>
>
> ### Ad. W2: Why BV Merging Is Practical and Advantageous for the Adversary
>
> - The key advantage of formulating backdoor attacks as BVs is **their transferability**:  an adversary can take a single BV and **add it to many benign task vectors** to obtain effective backdoor models *without any retraining*.
>
> - A second advantage is the ability to create a **non-interfering and stronger final attack**.  Because BVs are sparse and nearly orthogonal, merging multiple BVs yields a **more potent and more resilient multi-backdoor attack** than any individual backdoor. We further show that the BV formulation enables the construction of **SBV attacks that are significantly stronger**, forming a Pareto frontier in Fig. 4 and Fig. 5.
>
> - Finally, **training multiple single-purpose backdoors is simpler, more stable, and more stealthy** than training a single large multi-trigger model, which often suffers from interference and accuracy degradation.  With BVs, each backdoor can be trained independently, and the adversary can freely merge any subset afterward **without joint optimization or additional training cost**.
>
> ### Ad. W3: Experimental Evidence Supporting Our Claims
> We emphasize that our main goal is to introduce the BV framework and demonstrate its practical value. Our evaluation was designed around this objective and was already extensive, spanning 21 pages with 15 tables and 13 figures.
>
> During the rebuttal, we further expanded our experiments by adding three additional backdoor attacks (WaNet, Blend, BadTV) to SBV and IBVS evaluations, introducing adaptive attacks on IBVS, and analyzing BV’s components in detail.
>
> Across all settings, results consistently show that **SBV attack is much stronger and more resilient when merging multiple models**, and **IBVS suppresses unknown triggers**, confirming the practical usefulness of the BV framework. In a comment below we summarize the experiments from the main text and appendix, grouped according to their role in presenting the BV framework (see points 6–8 for IBVS details).
>
> We go into details in additional comments to address the rest of specific concerns raised and present results of additional evaluation.
>
> ### Final remarks
>
> **We sincerely appreciate Reviewer's constructive feedback, which has helped us further improve our work.** If the Reviewer has any additional questions, we would be happy to address them.
>
> If our responses have resolved the concerns, we kindly ask the Reviewer to consider increasing the initial score.

---

> > ### Author Response · Authors · 2025-11-25
> > **Details for answers to W1 and W3**
> >
> > ### Ad. W1: Comparison of our work to BadTV - details
> >
> > | **Aspect** | **BadTV(Arxiv,2025)** | **Backdoor Vectors(ours)** |
> > |-----------|------------------------|----------------------------|
> > | **1. Contribution** | First backdoor attack that stays effective under **addition, subtraction, and task analogy** - traditional attacks fail under subtraction. | First framework to treat **backdoors as task vectors**, enabling vector-space analysis, backdoor interference (transfer) measurement, BV merging and a unified attack/defense view. |
> > | **2. Main goal** | Show that task vectors in TA/TVaaS can be compromised and introduce a robust composite backdoor (BadTV). | Provide a vector-based formulation of backdoors and show how this framework can be used to better understand backdoors in MM. |
> > | **3. Use of task vectors (definition)** | Specific attack: **BadTV = α₁·b1 − α₂·b2**, where b1, b2 are backdoors; b1 activates under addition and b2 under subtraction; built via asymmetric **backdoor fine-tuning**. | More general: BV = **θ_backdoored − θ_clean**. Adding BV injects a backdoor; subtracting removes it. We may represent multiple attacks in our general BV framework, including BadTV. |
> > | **4. Attack/defense** | Attack-focused. | Backdoor-focused; provides both **attack (SBV)** and **defense (IBVS)** to show the vector-based framework is useful. |
> > | **5. Metrics** | Clean Accuracy(CA),Attack Success Rate(ASR). | CA, ASR, Backdoored Accuracy(BA), BV sparsity & BV cosine similarity. BV formulation captures similarity and transfer of different backdoor attacks. |
> > | **6. MM setup** | multi-task | mostly single-task, additional experiments for multi-task |
> > ||||
> >
> >
> > ### Ad. W3: Experimental Evidence Supporting Our Claims - details
> >
> > While no empirical evaluation can be entirely exhaustive, we believe the scope and depth of our experiments are sufficient to clearly establish the practical value of representing backdoors as task vectors.
> >
> > To this end, we propose the BV formulation and conduct a comprehensive empirical study demonstrating its usefulness:
> >
> > 1. **Backdoor behavior under BV addition**:  We show that adding a BV preserves clean accuracy while increasing ASR (Fig. 2).
> >
> > 2. **Backdoor interference and transferability**:  We measure how backdoors interact across different optimization seeds, trigger locations, targets, trigger sizes, attack types, tasks, and base models (Fig. 3, Fig. 9, Fig. 10, Fig. 11, Fig. 13).
> >
> > 3. **BV similarity and sparsity analysis**: Comparing backdoor attacks beyond ASR metric (Tab. 1, Tab. 2)
> >
> > 4. **Strength of merged backdoors (SBV)**:  We demonstrate that merging multiple BVs yields a more resilient and powerful attack (Tab. 3, Fig. 4, Fig. 5), validated across single- and multi-task settings (Tab. 9), different base models (ViT-B, ViT-L, ConvNeXt; Tab. 12, Tab. 14), and different merging rules and TV merging methods such as TA and TIES (Tab. 3, Tab. 11).
> >
> > 5. **BV insights: discovering the role of inherent triggers in model merging attacks**: We show that inherent backdoors are far more influential than injected ones in MM settings (Fig. 7, Fig. 8, Fig. 12).
> >
> > 6. **IBVS defense leveraging BV similarity**:  We propose IBVS as a simple yet effective baseline defense (Fig. 5), and evaluate it across  BadNets triggers (Tab. 4), backbone models (ViT-B, ViT-L, ConvNeXt; Tab. 13, Tab. 15), and BV merging methods (Tab. 10).
> >
> > 7. **Additional ablations**:  We include detailed analyses of SBV strength (Tab. 7) and IBVS performance across different *k* values (Tab. 8).
> >
> > 8. **Rebuttal-period extensions**:  We further expanded our evaluation with three additional backdoor attacks - **WaNet, Blend, and BadTV** - including full SBV and IBVS evaluations, adaptive attacks on IBVS, and additional BV component analysis.

---

> > > ### Author Response · Authors · 2025-11-25
> > > **SBV/IBVS evaluation for WaNet, Blend, BadTV attacks**
> > >
> > > ### W4 SBV/IBVS evaluation for additional backdoor attacks
> > >
> > > We extended our evaluation with three backdoor methods:
> > > - **WaNet** – A warping-based backdoor that applies subtle, smooth geometric distortions to the whole image, not a visible patch trigger.
> > > - **Blend** – A full-image blending backdoor where a trigger pattern is softly mixed with the entire image, so no specific patch is required.
> > > - **BadTV** – A task-vector–based composite backdoor created using TV operations (addition/subtraction of task vectors) to generate a more robust backdoor.
> > >
> > > The base experimental setup is the same as reported in Tab. 3 and Tab. 4 (ViT-B, 10 tasks, merging coef=0.1).
> > >
> > > ### SBV performance
> > >
> > > SBV attacks demonstrate strong scalability to diverse backdoor types. Crucially, merging several sparse backdoor vectors consistently enhances the attack's potency, a benefit that persists even when the underlying backdoor mechanism is altered.
> > >
> > > | **Attack** | **Variant** | **CIFAR100 BA** | **CIFAR100 ASR** | **ImageNet100 BA** | **ImageNet100 ASR** |
> > > |-----------|-------------|------------------|-------------------|---------------------|----------------------|
> > > | **Blend** | **BV (no backdoor merging)** | 90.04 ± 0.06 | 1.81 ± 2.64 | 86.25 ± 0.12 | 0.44 ± 0.56 |
> > > |           | **SBV (ours)** | 89.54 ± 0.08 | **97.54 ± 0.53** | 86.14 ± 0.08 | **55.42 ± 4.41** |
> > > |           | **Δ (SBV − BV)** | −0.50 | **+95.73** | −0.11 | **+54.98** |
> > > | **WaNet** | **BV (no backdoor merging)** | 89.90 ± 0.05 | 0.17 ± 0.12 | 86.22 ± 0.06 | 0.31 ± 0.30 |
> > > |           | **SBV (ours)** | 89.54 ± 0.11 | **39.32 ± 6.29** | 85.94 ± 0.25 | **66.15 ± 3.98** |
> > > |           | **Δ (SBV − BV)** | −0.36 | **+39.15** | −0.28 | **+65.84** |
> > > | **BadTV** | **BV (no backdoor merging)** | 89.72 ± 0.05 | 0.18 ± 0.14 | 85.75 ± 0.09 | 0.34 ± 0.21 |
> > > |           | **SBV (ours)** | 88.66 ± 0.16 | **11.22 ± 5.37** | 83.94 ± 0.35 | **12.11 ± 3.95** |
> > > |           | **Δ (SBV − BV)** | −1.06 | **+11.04** | −1.81 | **+11.77** |
> > > |||||||
> > >
> > > ### IBVS results for SBV attacks
> > >
> > > IBVS strongly reduces SBV-WaNet and SBV-BadTV attacks, while it provides only limited reduction for SBV-Blend. We use BadNets wavelet triggers for IBVS as in row 3 in Tab. 4.
> > > |**Attack**|**λ₁**|**CIFAR100 BA**|**CIFAR100 ASR**|**ImageNet100 BA**|**ImageNet100 ASR**|
> > > |----------|-----|----------------|-----------------|-------------------|--------------------|
> > > |**Blend**|**0.1**|89.52|97.44|86.10|54.60|
> > > |         |Δ vs 0.0|−0.02|**−0.10**|−0.04|**−0.82**|
> > > |         |**0.3**|89.40|97.14|86.02|52.77|
> > > |         |Δ vs 0.0|−0.14|**−0.40**|−0.12|**−2.65**|
> > > |**WaNet**|**0.1**|89.52|38.23|85.94|61.90|
> > > |         |Δ vs 0.0|−0.02|**−1.09**|0.00|**−4.25**|
> > > |         |**0.3**|89.32|33.91|85.85|44.27|
> > > |         |Δ vs 0.0|−0.22|**−5.41**|−0.09|**−21.88**|
> > > |**BadTV**|**0.1**|88.63±0.20|8.66±4.08|83.90±0.36|11.05±3.40|
> > > |         |Δ vs 0.0|−0.03|**−2.56**|−0.04|**−1.06**|
> > > |         |**0.3**|88.49±0.21|5.29±2.66|83.78±0.32|9.15±2.67|
> > > |         |Δ vs 0.0|−0.17|**−5.93**|−0.16|**−2.96**|

---

> > > > ### Author Response · Authors · 2025-11-25
> > > > **Other clarifications**
> > > >
> > > > *Q2 What is the target class (alone) accuracy on clean images after subtracting backdoor vectors?*
> > > >
> > > > We present those values below as the extension of Tab 4:
> > > >
> > > > | **Lambda 1**            | **CIFAR100 BA (BV / SBV)** | **CIFAR100 target Acc (BV / SBV)** | **CIFAR100 ASR (BV / SBV)** | **ImageNet100 BA (BV / SBV)** | **ImageNet100 target Acc (BV / SBV)** | **ImageNet100 ASR (BV / SBV)** |
> > > > |-------------------------|------------------------------|------------------------------------|-------------------------------|--------------------------------|----------------------------------------|--------------------------------|
> > > > | **λ₁ = 0.0 (no defense)** | 89.74 / 89.69               | 89.20 / 89.00                       | 27.29 / 97.21                | 85.91 / 85.83                  | 87.20 / 87.60                           | 84.01 / 99.99                  |
> > > > | **λ₁ = 0.1**             | 89.77 / 89.66               | 89.20 / 89.40                       | 23.55 / 95.45                | 85.88 / 85.79                  | 87.60 / 87.60                           | 78.59 / 99.97                  |
> > > > | **Δ to λ₁ = 0**          | +0.03 / −0.03               | **0.00 / +0.40**                    | −3.74 / −1.76               | −0.03 / −0.04                 | **+0.40 / 0.00**                        | −5.42 / −0.02                 |
> > > > | **λ₁ = 0.3**             | 89.57 / 89.46               | 89.80 / 89.00                       | 16.56 / 89.21                | 85.61 / 85.59                  | 89.60 / 87.60                           | 65.26 / 99.89                  |
> > > > | **Δ to λ₁ = 0**          | −0.17 / −0.23               | **+0.60 / 0.00**                    | −10.73 / −7.99              | −0.30 / −0.24                 | **+2.40 / 0.00**                        | −18.75 / −0.10                |
> > > >
> > > >
> > > > *Q3. In Table 3, does it mean merging is better than not merging? Is the bad merging ASR lower than before merging (27%)?*
> > > >
> > > > In Tab. 3 we report the ASR after merging 10 models, one backdoored, nine clean (benign). Proposed BV merging (SBV) is much better (**+69.92pp** CIFAR100, **+15.98pp** ImageNet100) than a single BadMerging attack (reported under BV row with ASR=27% CIFAR100, ASR=84% for ImageNet100).
> > > >
> > > > *Q4. In Table 4, are the results for the same badnets with different triggers?*
> > > >
> > > > In Tab. 4, we report IBVS results where the defense uses two different BadNets triggers: the defensive BV is built either from the white-square trigger (row 2) or from the wavelet trigger (row 3).
> > > >
> > > > *Q5. The paper says addition and subtraction of BVs are not symmetric. Then what are the guarantees of the backdoor defense (IBVS)?*
> > > >
> > > > We do **not** claim that task-vector addition or subtraction is asymmetric.
> > > >
> > > > What we state is that backdoor transfer - as defined in our BV section using ASR as the metric — is asymmetric (Fig. 9). We observe two forms of asymmetry:
> > > > - **From inherent-trigger attacks to injected-trigger attacks**: injected triggers transfer more strongly into inherent (in ASR terms) than the other way around.
> > > > - **From small trigger patches to larger ones**: small triggers influence larger-trigger attacks more than the reverse.
> > > >
> > > > Since in IBVS we subtract injection triggers for defense (for the  difference between injected triggers vs inherent in MM - see sec 4.4), **the first asymmetry actually benefits IBVS**.
> > > >
> > > > Subtracting a BV associated with a smaller trigger than in the attack can weaken IBVS. This is evident in our Blend attack results, where using a naive small-patch IBVS pattern (1% of image) is much less effective when defending against full-image blend triggers.
> > > >
> > > > *Q6. In Table 9, the adversary tasks from the caption are different from the ones inside the table. How do the multi-tasks work for the results?*
> > > >
> > > > The caption of Fig. 9 conveys the following: in the multi-task setting, we always merge six models. The first model corresponds to the adversary’s task (either CIFAR-100 or ImageNet-100), while the remaining five are clean models fine-tuned on the datasets listed in the caption. Thus, multi-task merging always involves six models in total, with the adversary’s task serving as the first one.
> > > >
> > > > *The shorthands, SC, RND are not clearly defined in the paper for Algorithm 1, although the reviewer understands them as sign consistent and random.*
> > > >
> > > > Correct.

---

### Official Review · Reviewer_w4na · 2025-10-29

**Soundness:** 4
**Presentation:** 3
**Contribution:** 3
**Rating:** 8
**Confidence:** 3

**Summary:**

This paper introduces Backdoor Vectors (BVs) as a task-arithmetic-based framework to analyze and mitigate backdoor attacks in model merging (MM). By interpreting backdoor attacks as task vectors, the authors identify key properties of attack transferability and propose two main contributions: (1) Sparse Backdoor Vector (SBV) – a method to merge multiple BVs into a more resilient backdoor, and (2) Injection BV Subtraction (IBVS) – a simple, assumption-free defense against unknown backdoors. Experiments on CLIP-based models and various datasets show that SBV strengthens attack persistence, while IBVS effectively mitigates unseen threats.

**Strengths:**

- The discussion of backdoor learning in model merging is new and meaningful. This paper can provide insights to inspire future work in this field.
- The work presents a novel conceptual framing of backdoor attacks as “task arithmetic,” providing a unified analytical view that bridges attack and defense perspectives.
- The paper includes solid empirical validation with diverse datasets (CIFAR100, ImageNet100, TinyImageNet) and architectures (ViT, ConvNext), demonstrating effectiveness and robustness.
- The paper is well-structured with clear mathematical formulation and illustrative figures.

**Weaknesses:**

- The framework assumes linearity in task vectors, which may not hold in nonlinear regimes or across architectures, limiting the theoretical generality.
- While IBVS is lightweight, its robustness under adaptive or multi-stage attacks is not deeply explored; more comparison with recent adversarial defenses would strengthen the claims.
- Some sections (e.g., Section 3) are heavy on newly introduced terms (BV, SBV, BV⊕, BV⋆), which may hinder accessibility for readers outside the MM subcommunity.

**Questions:**

Could the proposed BV framework be extended to language or multi-modal models beyond CLIP?

---

> ### Author Response · Authors · 2025-11-26
>
> We thank the Reviewer for their thoughtful feedback and are pleased that they view our contribution to model merging security as new and meaningful, with the potential to inspire future work in this emerging area.
>
> We are especially pleased that the Reviewer appreciates the **solid empirical validation** of our claims and recognizes the value of the **unified perspective on backdoors that our framework offers**.
>
> We would like to address three specific concerns and one question raised by the Reviewer:
>
> **W1)** A question regarding our method’s limitation to regimes in which the linear-addition assumption holds.
> **W2)** The absence of an adaptive attack against the proposed IBVS method.
> **W3)** The potential challenge that newly introduced symbols may pose for some readers.
> **Q1)** Whether our BV framework can be applied beyond CLIP models.
>
>
> ### W1.  On the assumption of linear addition in MM
> Our work builds on the hypothesis that backdoor vectors are linearly additive. This assumption is standard in the model-merging literature, and we clearly acknowledge it in our limitations section.
>
> Multiple works, such as [1], identify that linear addition is effective when operating in a linear connectivity regime, which is induced by the scale of the model and amount of pretraining. This is why we perform our experiments on relatively large vision models (ViT-B and ViT-L, ConvNext) pretrained with image-caption contrastive pretraining (CLIP) on a massive dataset.
>
> To the best of our knowledge, current MM research primarily - if not exclusively - focuses on merging models that share the same architecture. While our findings may not directly extend to future cross-architecture merging scenarios or to regimes where linear additivity does not hold, **this does not pose a substantial limitation within the scope of present MM practice**.
>
> [1] Yadav et al. What Matters for Model Merging at Scale?, arXiv preprint 2024
>
> ### W2.  Adaptive Attack against IBVS defense - additional results
>
> IBVS demonstrates how the BV framework can be applied in practice and follows naturally from the observations revealed by our BV analysis in model merging. We therefore recognize this concern - raised by multiple Reviewers - as important and, in response, present an adaptive attack on IBVS.
>
> The proposed adaptive attack increases ASR slightly in the no-defense setting, but does not overpower IBVS when the defense is present, confirming that **IBVS is effective even against adaptive attackers with full knowledge of the trigger and dataset**.
>
> ### W3. Improving accessibility for readers outside the MM community
> We will add a table that clearly lists and explains all terms and operators (e.g., BV, SBV, BV⊕, BV⋆) introduced and used throughout the paper. We believe this addition will make the presentation more approachable and improve overall readability.
>
> ### Q1. Whether our BV framework can be applied beyond CLIP models.
> The short answer is: yes.
>
> The applicability of the BV framework is broad, owing to its general formulation. For example, in our related work, we discuss several MM studies involving backdoors in LLMs, and many of our observations align closely with those findings. Some of these observations could also help illuminate LLM-based model-merging phenomena at a deeper level, offering additional structure and interpretability to behaviors reported in recent LLM MM studies.
>
> This suggests that BVs may serve as a **useful lens for understanding, comparing, detecting, and mitigating backdoors and even biases in the language domain**. In particular, we hope that BV analysis may help explain or predict forms of misalignment in fine-tuned models, as illustrated by recent work that has attracted significant attention within the AI safety community [1]. We aim to analyze those aspects in depth in our future research.
>
> [1] Emergent Misalignment: Narrow finetuning can produce broadly misaligned LLMs, ICML 2025
>
> ### Final Remarks
>
> We sincerely thank the Reviewer for their time and effort, and we are glad they appreciate the novelty of our contribution, the unified perspective on backdoors, the strong empirical validation across diverse settings, and the clarity of our presentation.
>
> We hope that our responses have adequately addressed the concerns raised, and we remain available to provide any further clarification or answer additional questions to increase the Reviewer’s confidence score.

---

> > ### Author Response · Authors · 2025-11-26
> > **W2. Adaptive Attack against IBVS defense - additional results**
> >
> > #### Assumptions
> >
> > - **(A1)** The adversary knows the trigger and dataset that will be used for the IBVS defense.
> > - **(A2)** The adversary may backdoor **only a single model** (as in our setting).
> > - **(A3)** The defender chooses:
> >   - **λ** — the merging coefficient for combining multiple models,
> >   - **λ₁** — the scaling coefficient for subtracting the IBVS defense vector.
> > - **(A4)** The adversary chooses **λ₂**, the scaling coefficient for adding their adaptive vector **BV′**.
> >
> > ---
> >
> > #### Attack Scenario
> >
> > 1. The adversary constructs an **adaptive BV′** using the known trigger and dataset.
> > 2. To counteract the expected defensive BV (IBVS), the adversary **adds additional BV′ to their backdoored model**, scaled by **λ₂**.
> >
> > ---
> >
> > #### Empirical Findings
> > Simple convention: (AA - Adaptive Attack/ IBVS) - ticks when AA or IBVS are present: (✓ / ✗) means AA present, IBVS defense not present.
> >
> > Our evaluation of this adaptive attack yields several insights:
> > 1. **Adding BV′ creates a multi-trigger attack**, effectively implanting two backdoors (e.g., BadMerging + BadNets).
> > 2. **Without IBVS defense**, (✓ / ✗) adding BV′ slightly **increases ASR** for the main attack.
> > 3. **With IBVS**, (✓ / ✓) BV′ can **partially reduce the defense’s effectiveness** (compared to ✗ / ✓ ), but usually does not fully negate it.
> > 4. **The adversary typically cannot overpower IBVS** when contributing only one backdoored model. Their influence is limited by  **λ₂** * **λ** < **λ₁** * **λ** * **#models_merged** meaning the defender’s cumulative IBVS effect grows with model count, while the adversary’s impact remains bounded.
> >
> >
> > ### BV results
> > We present the extension of Tab. 4 (merging 10 models with merging coeficient 0.1) for adaptive attacks (AA) with BadNets' white square trigger used by AA and IBVS.
> > (AA - Adaptive Attack, IBVS - if IBVS defense is present)
> >
> > **Without IBVS defense**, (✓ / ✗) adversary slightly **increases ASR** for the main attack using AA. **With IBVS**, (✓ / ✓) AA can **partially reduce the defense’s effectiveness** (compared to ✗ / ✓ ), but does not fully negate it.
> >
> >
> > | **AA / IBVS** | **λ₁/λ₂** | **λ** | **CIFAR100 CA/ASR** | **Δ CA / Δ ASR** | **ImageNet100 CA/ASR** | **Δ CA / Δ ASR** |
> > |------------------------|-----------|------------|------------------|------------------|----------------------|------------------|
> > | ✗ / ✗                  | 0.0 / 0.0 | 0.1        | 89.74 / 27.29    | 0.00 / 0.00      | 85.91 / 84.01        | 0.00 / 0.00      |
> > | ✓ / ✗                  | 0.0 / 1.0 | 0.1        | 89.70 / 28.44    | -0.04 / +1.15    | 85.94 / 87.71        | +0.03 / +3.70    |
> > | ✓ / ✗                  | 0.0 / 0.5 | 0.1        | 89.71 / 27.89    | -0.03 / +0.60    | 85.88 / 85.98        | -0.03 / +1.97    |
> > | ✓ / ✓                  | 0.1 / 1.0 | 0.1        | 89.71 / 27.63    | -0.03 / +0.34    | 85.88 / 84.88        | -0.03 / +0.87    |
> > | ✓ / ✓                  | 0.1 / 0.5 | 0.1        | 89.78 / 26.98    | +0.04 / -0.31    | 85.93 / 82.81        | +0.02 / -1.20    |
> > | ✗ / ✓                  | 0.1 / 0.0 | 0.1        | 89.82 / 25.60    | +0.08 / -1.69    | 85.93 / 78.84        | +0.02 / -5.17    |
> > | ✓ / ✓                  | 0.3 / 1.0 | 0.1        | 89.70 / 25.14    | -0.04 / -2.15    | 85.66 / 77.25        | -0.25 / -6.76    |
> > | ✓ / ✓                  | 0.3 / 0.5 | 0.1        | 89.70 / 24.16    | -0.04 / -3.13    | 85.61 / 74.62        | -0.30 / -9.39    |
> > | ✗ / ✓                  | 0.3 / 0.0 | 0.1        | 89.58 / 20.65    | -0.16 / -6.64    | 85.53 / 66.83        | -0.38 / -17.18   |
> >
> >
> > ---
> > ### SBV results
> >
> > Same insights apply for SBV attacks, but SBV attacks are much more robust to proposed defense.
> >
> > | **AA / IBVS** | **λ₁/λ₂** | **λ** | **CIFAR100 CA/ASR** | **Δ CA / Δ ASR** | **ImageNet100 CA/ASR** | **Δ CA / Δ ASR** |
> > |------------------------|-----------|------------|------------------|------------------|----------------------|------------------|
> > | ✗ / ✗  | 0.0 / 0.0 | 0.1        | 89.69 / 97.21    | 0.00 / 0.00      | 85.83 / 99.99        | 0.00 / 0.00      |
> > | ✓ / ✗    | 0.0 / 1.0 | 0.1        | 89.68 / 97.37    | -0.01 / +0.16    | 85.84 / 100.00       | +0.01 / +0.01    |
> > | ✓ / ✗    | 0.0 / 0.5 | 0.1        | 89.65 / 97.32    | -0.04 / +0.11    | 85.85 / 100.00       | +0.02 / +0.01    |
> > | ✓ / ✓       | 0.1 / 1.0 | 0.1        | 89.72 / 97.00    | +0.03 / -0.21    | 85.80 / 99.99        | -0.03 / 0.00     |
> > | ✓ / ✓  | 0.1 / 0.5 | 0.1        | 89.74 / 96.85    | +0.05 / -0.36    | 85.84 / 99.98        | +0.01 / -0.01    |
> > | ✗ / ✓    | 0.1 / 0.0 | 0.1        | 89.69 / 96.75    | 0.00 / -0.46     | 85.77 / 99.96        | -0.06 / -0.03    |
> > | ✓ / ✓  | 0.3 / 1.0 | 0.1        | 89.61 / 95.19    | -0.08 / -2.02    | 85.57 / 99.96        | -0.26 / -0.03    |
> > | ✓ / ✓    | 0.3 / 0.5 | 0.1        | 89.63 / 94.74    | -0.06 / -2.47    | 85.53 / 99.94        | -0.30 / -0.05    |
> > | ✗ / ✓       | 0.3 / 0.0 | 0.1        | 89.61 / 94.69    | -0.08 / -2.52    | 85.47 / 99.83        | -0.36 / -0.16    |

---

### Official Review · Reviewer_Foyn · 2025-10-31

**Soundness:** 3
**Presentation:** 3
**Contribution:** 3
**Rating:** 4
**Confidence:** 4

**Summary:**

This paper proposes a novel perspective, modeling backdoor attacks as task vectors (BVs), and systematically analyzes the backdoor attack and defense mechanisms in model fusion. The authors propose two main methods:

- SBV (Sparse Backdoor Vector) improves attack robustness and success rate by merging multiple backdoor vectors through sparsification.

- IBVS (Injection BV Subtraction) is a lightweight defense method that suppresses unknown attacks by subtracting a fixed backdoor vector.

**Strengths:**

- Treating backdoor attacks as task vectors provides a novel perspective to understand backdoor behavior.

- The writing is easy to understand.

- The paper considers both attacking and defending.

**Weaknesses:**

- The assumption of linear additivity of task vectors is not explored in depth.

- The paper only includes comparisons with BadMerging and BadNets.

- No adaptive attack experiments against IBVS.

**Questions:**

The idea of BV does not sound reasonable, since there are only a small number of weights contributing to backdoor behaviors, which is proven by backdoor pruning works. However, BV contains a large number of weight variations related to the main task (clean behavior). This makes BV a mixture of a highly sparse "backdoor signal" and a dense "benign task noise".

It is possible that the benign weights are similar in both backdoor and benign models, so subtracting the weights of a clean fine-tuned model from those of a backdoored fine-tuned model could ignore the benign weights. But the authors should explicitly demonstrate it. I suggest that the authors provide some experiments to analyze why BV is effective rather than only showing the attacking performance.

---

> ### Author Response · Authors · 2025-11-25
>
> We thank the Reviewer for the valuable feedback. We are pleased that our Backdoor Vectors are seen as a novel, interesting perspective to understand backdoor behavior.
>
> Below, we address the key questions and concerns: first, we provide a brief overview of the most important points, followed by detailed responses in the additional comments. We identified four key concerns raised by the reviewer:
>
> - **Q1)** whether defining BV is necessary, since prior pruning-based works suggest that backdoor behavior may stem from a small subset of (high-magnitude) weights;
> - **W1)** the need for stronger theoretical justification of the linear additivity assumption;
> - **W2)** the limited number of backdoor attacks included in our comparison;
> - **W3)** the absence of an adaptive attack evaluation against IBVS.
> We address each of these concerns below:
>
> ### Q1: BV definition concern (grounded in backdoor pruning works).
>
> 1. We appreciate the reviewer’s thoughtful concern that expressing the backdoor as a BV might be superfluous. We view this as a valuable perspective and a useful opportunity to clarify the motivation for our BV formulation. Prior pruning works rely primarily on ASR, but **ASR captures only a single observable manifestation** of a backdoor. We argue, that it cannot fully characterize the underlying mechanisms. This underscores the need for more expressive representations - such as our **BV framework** - that captures the structure of backdoor signals beyond what ASR alone can reveal.
>
> 2. Raised concern is a direct motivation for the introduction of our  **sparse backdoor vectors (SBVs)**, which further constrains task information in BV. The sign-consistency heuristic makes BV sparser while simultaneously strengthening important backdoor weights.
>
> 3. Most pruning studies analyze backdoors created via data poisoning, whereas **backdoor signals introduced during training (as in our work) are more distributed across weights** than those inserted by data poisoning. We demonstrate this by analyzing BV for three settings: BadNets via data poisoning, BadNets via training poisoning, and BadMerging (training poisoning). Our BV framework generalizes across these settings while preserving their distinguishing characteristics.
>
> For a more detailed answer to Q1 see comment below.
>
> ### W1.  Theoretical background: assumption of linear addition in MM
>
> Our work indeed hypothesizes that backdoor vectors are linearly additive. This is a common assumption in model merging literature. Task Arithmetic [1] was one of the first works to edit the models by adding or removing various capabilities by linear addition or subtraction. Interestingly, they showed that linear additivity applies not only to narrow, well-defined tasks (like classification of digits) but also to broad properties of models such as toxicity of text generated by language models. We extended this observation to vulnerability to backdoor attacks.
>
> Multiple works, such as [2], identify that linear addition is effective when operating in a linear connectivity regime, which is induced by the scale of the model and amount of pretraining. This is why we perform our experiments on relatively large vision models (ViT-B and ViT-L, ConvNext) pretrained with image-caption contrastive pretraining (CLIP) on a massive dataset.
>
> The linear addition hypothesis is, to some extent, supported by our backdoor transfer experiments.
>
> [1] Ilharco et al. Editing Models with Task Arithmetic, ICLR 2023
>
> [2] Yadav et al. What Matters for Model Merging at Scale?, arXiv preprint 2024
>
> ### W2: SBV/IBVS evaluation for additional backdoor attacks
>
> We provide additional results for WaNet, Blend, and BadTV attacks below, which **further reinforce the empirical support for our findings.**
>
> ### W3 Adaptive attack on IBVS
>
> Although IBVS is not the central contribution of this paper, it illustrates how the BV framework can be applied in practice and emerges directly from the insights provided by our BV analysis in model merging. Thus, we recognize the Reviewer’ concern as important and present an adaptive attack on IBVS.
>
> Proposed adaptive attack increases ASR slightly in the no-defense setting, but do not overpower IBVS when the defense is present, confirming that **IBVS is effective even against adaptive attackers with full knowledge of the trigger and dataset**.
>
> ### Final Remarks
>
> We sincerely appreciate the Reviewer’s constructive feedback, which has helped us further strengthen the paper. We would be glad to address any remaining questions.
>
> If our clarifications resolve the raised concerns, we kindly ask the Reviewer to consider adjusting the initial score.

---

> > ### Author Response · Authors · 2025-11-25
> > **Q1: BV definition concern - details.**
> >
> > **Ad1:** While ASR is a widely used and intuitively simple metric (we also use it in our work!), it **captures only one observable manifestation** of a backdoor, and does not fully reflect the underlying representational or structural changes inside the model. Therefore, it offers an inherently partial view of the phenomenon. One of the core motivations behind the BV framework is precisely to represent **all information associated with backdoor behavior**, including the portion that is intertwined with the task - since, in practice, backdooring necessarily leverages specific representations and data. We argue that BV provides a **general and concrete characterization** of backdoors that can be quantitatively analyzed, even in cases where the backdoor manifests only as a representational bias rather than a strong, easily measurable ASR signal. In this sense, our framework constitutes a **first and essential step toward a more complete understanding** of backdoors in MM in their full complexity.
> >
> > **Ad2:** From another perspective the reviewer's observation is a strong motivation for our proposed sparse backdoor vectors attack method, where backdoor information is strengthened in a smaller number of weights, and non-backdoor information is explicitly removed with (sign-consistency) heuristic. Although future heuristics may improve this, we show that even this simple approach yields a much stronger merged attack (in ASR) by explicitly removing less relevant BV components. To show the difference between BV and SBV we present the BV pruning procedure below:
> >
> >
> >
> > ### ASR Under BV and SBV Pruning
> >
> > **Pruning Procedure Explanation**
> >
> > In our pruning procedure, **x%** denotes the fraction of BV weights with the **smallest magnitudes** that are *retained*, while all remaining (larger-magnitude) weights are removed.
> > Thus, higher x% values correspond to keeping **more** BV weights, gradually reintroducing parameters with increasingly larger magnitudes. This provides a controlled way to test whether backdoor behavior relies primarily on **large-magnitude BV parameters**, i.e., weights that differ most strongly between the backdoored and clean models.
> >
> >
> > BM - BadMerging, BVs are for CIFAR100.
> >
> > | **Percentage kept (x%)** | 0     | 50    | 55    | 60    | 65    | 70    | 75    | 80    | 85     | 90     |
> > |--------------------------|-------|-------|-------|-------|-------|-------|-------|-------|--------|--------|
> > | **BV (BM)**             | 10.02 | 15.25 | 20.36 | 28.22 | 41.09 | 58.97 | 79.72 | 94.96 | 99.76  | 100.00 |
> > | **SBV (BM)**            | 10.02 | 10.02 | 10.02 | 10.02 | 10.02 | 10.02 | 22.40 | 87.71 | 100.00 | 100.00 |
> >
> > SBV effectively amplifies the strong backdoor components while removing the weak backdoor signal contained in the small-magnitude weights of the BV derived from the BadMerging attack.
> >
> >
> > **Ad3:** Most pruning works show results for data poisoning with backdoor samples (as a most general threat). In model merging there exists an additional threat of adding a backdoor differently: in this work, following BadMerging we add backdoors during training procedure (as mentioned in RW). As a consequence, the backdoor signal is scattered more broadly than in data-poisoning backdoor approaches.
> >
> > ### ASR Under BV Pruning
> > *(x% = fraction of smallest-magnitude BV weights retained)*
> > BN - BadNets, BM - BadMerging.
> >
> > | **Percentage kept (x%)** | 0     | 50    | 55    | 60    | 65    | 70    | 75    | 80    | 85     | 90      |
> > |--------------------------|-------|-------|-------|-------|-------|-------|-------|-------|--------|---------|
> > | **BV_data_poison_BN (ASR)** |   –   | 0.23  | 0.27  | 0.32  | 0.39  | 0.54  | 0.75  | 1.37  | 4.87   | 57.19   |
> > | **BV_train_poison_BN (ASR)**| 0.25  | 0.19  | 0.19  | 0.21  | 0.22  | 0.26  | 0.41  | 2.46  | 42.40  | 99.82   |
> > | **BV_train_poison_BM (ASR)**| 10.02 | 15.25 | 20.36 | 28.22 | 41.09 | 58.97 | 79.72 | 94.96 | 99.76  | 100.00  |
> >
> > By BV pruning we show that:
> > - **Backdoor Signal in BV is distributed and strong even in smaller-magnitude weights.**
> > - Backdoor signal is strong in BV, even after pruning 10% of the highest-magnitude weights.
> > - The distribution of backdoor signal in small weights is stronger when backdoor is implanted during training compared to bacdooring via data poisoning (difference between first and second row - the same attack).
> >
> > Our BV framework generalizes across different bacdoor implanting settings while preserving their distinguishing characteristics.

---

> > > ### Author Response · Authors · 2025-11-25
> > > **W2. SBV/IBVS evaluation for additional backdoor attacks**
> > >
> > > ### W2 SBV/IBVS evaluation for additional backdoor attacks
> > >
> > > We extended our evaluation with three backdoor methods:
> > > - **WaNet** – A warping-based backdoor that applies subtle, smooth geometric distortions to the whole image, not a visible patch trigger.
> > > - **Blend** – A full-image blending backdoor where a trigger pattern is softly mixed with the entire image, so no specific patch is required.
> > > - **BadTV** – A task-vector–based composite backdoor created using TV operations (addition/subtraction of task vectors) to generate a more robust backdoor.
> > >
> > > The base experimental setup is the same as reported in Tab. 3 and Tab. 4 (ViT-B, 10 tasks, merging coef=0.1).
> > >
> > > ### SBV performance
> > >
> > > SBV attacks demonstrate strong scalability to diverse backdoor types. Crucially, merging several sparse backdoor vectors consistently enhances the attack's potency, a benefit that persists even when the underlying backdoor mechanism is altered.
> > >
> > > | **Attack** | **Variant** | **CIFAR100 BA** | **CIFAR100 ASR** | **ImageNet100 BA** | **ImageNet100 ASR** |
> > > |-----------|-------------|------------------|-------------------|---------------------|----------------------|
> > > | **Blend** | **BV (no backdoor merging)** | 90.04 ± 0.06 | 1.81 ± 2.64 | 86.25 ± 0.12 | 0.44 ± 0.56 |
> > > |           | **SBV (ours)** | 89.54 ± 0.08 | **97.54 ± 0.53** | 86.14 ± 0.08 | **55.42 ± 4.41** |
> > > |           | **Δ (SBV − BV)** | −0.50 | **+95.73** | −0.11 | **+54.98** |
> > > | **WaNet** | **BV (no backdoor merging)** | 89.90 ± 0.05 | 0.17 ± 0.12 | 86.22 ± 0.06 | 0.31 ± 0.30 |
> > > |           | **SBV (ours)** | 89.54 ± 0.11 | **39.32 ± 6.29** | 85.94 ± 0.25 | **66.15 ± 3.98** |
> > > |           | **Δ (SBV − BV)** | −0.36 | **+39.15** | −0.28 | **+65.84** |
> > > | **BadTV** | **BV (no backdoor merging)** | 89.72 ± 0.05 | 0.18 ± 0.14 | 85.75 ± 0.09 | 0.34 ± 0.21 |
> > > |           | **SBV (ours)** | 88.66 ± 0.16 | **11.22 ± 5.37** | 83.94 ± 0.35 | **12.11 ± 3.95** |
> > > |           | **Δ (SBV − BV)** | −1.06 | **+11.04** | −1.81 | **+11.77** |
> > > |||||||
> > >
> > > ### IBVS results for SBV attacks
> > >
> > > IBVS strongly reduces SBV-WaNet and SBV-BadTV attacks, while it provides only limited reduction for SBV-Blend. We use BadNets wavelet triggers for IBVS as in row 3 in Tab. 4.
> > > |**Attack**|**λ₁**|**CIFAR100 BA**|**CIFAR100 ASR**|**ImageNet100 BA**|**ImageNet100 ASR**|
> > > |----------|-----|----------------|-----------------|-------------------|--------------------|
> > > |**Blend**|**0.1**|89.52|97.44|86.10|54.60|
> > > |         |Δ vs 0.0|−0.02|**−0.10**|−0.04|**−0.82**|
> > > |         |**0.3**|89.40|97.14|86.02|52.77|
> > > |         |Δ vs 0.0|−0.14|**−0.40**|−0.12|**−2.65**|
> > > |**WaNet**|**0.1**|89.52|38.23|85.94|61.90|
> > > |         |Δ vs 0.0|−0.02|**−1.09**|0.00|**−4.25**|
> > > |         |**0.3**|89.32|33.91|85.85|44.27|
> > > |         |Δ vs 0.0|−0.22|**−5.41**|−0.09|**−21.88**|
> > > |**BadTV**|**0.1**|88.63±0.20|8.66±4.08|83.90±0.36|11.05±3.40|
> > > |         |Δ vs 0.0|−0.03|**−2.56**|−0.04|**−1.06**|
> > > |         |**0.3**|88.49±0.21|5.29±2.66|83.78±0.32|9.15±2.67|
> > > |         |Δ vs 0.0|−0.17|**−5.93**|−0.16|**−2.96**|

---

> > > > ### Author Response · Authors · 2025-11-25
> > > > **W3. Adaptive Attack against IBVS**
> > > >
> > > > #### Assumptions
> > > >
> > > > - **(A1)** The adversary knows the trigger and dataset that will be used for the IBVS defense.
> > > > - **(A2)** The adversary may backdoor **only a single model** (as in our setting).
> > > > - **(A3)** The defender chooses:
> > > >   - **λ** - the merging coefficient for combining multiple models,
> > > >   - **λ₁** - the scaling coefficient for subtracting the IBVS defense vector.
> > > > - **(A4)** The adversary chooses **λ₂**, the scaling coefficient for adding their adaptive vector **BV′**.
> > > >
> > > > ---
> > > >
> > > > #### Attack Scenario
> > > >
> > > > 1. The adversary constructs an **adaptive BV′** using the known trigger and dataset.
> > > > 2. To counteract the expected defensive BV (IBVS), the adversary **adds additional BV′ to their backdoored model**, scaled by **λ₂**.
> > > >
> > > > ---
> > > >
> > > > #### Empirical Findings
> > > > Simple convention: (AA - Adaptive Attack/ IBVS) - ticks when AA or IBVS are present: (✓ / ✗) means AA present, IBVS defense not present.
> > > >
> > > > Our evaluation of this adaptive attack yields several insights:
> > > > 1. **Adding BV′ creates a multi-trigger attack**, effectively implanting two backdoors (e.g., BadMerging + BadNets).
> > > > 2. **Without IBVS defense**, (✓ / ✗) adding BV′ slightly **increases ASR** for the main attack.
> > > > 3. **With IBVS**, (✓ / ✓) BV′ can **partially reduce the defense’s effectiveness** (compared to ✗ / ✓ ), but usually does not fully negate it.
> > > > 4. **The adversary typically cannot overpower IBVS** when contributing only one backdoored model. Their influence is limited by  **λ₂** * **λ** < **λ₁** * **λ** * **#models_merged**, meaning the defender’s cumulative IBVS effect grows with model count, while the adversary’s impact remains bounded.
> > > >
> > > >
> > > > ### BV results
> > > > We present the extension of Tab. 4 (merging 10 models with merging coeficient 0.1) for adaptive attacks (AA) with BadNets' white square trigger used by AA and IBVS.
> > > > (AA - Adaptive Attack, IBVS - if IBVS defense is present)
> > > >
> > > > **Without IBVS defense**, (✓ / ✗) adversary slightly **increases ASR** for the main attack using AA. **With IBVS**, (✓ / ✓) AA can **partially reduce the defense’s effectiveness** (compared to ✗ / ✓ ), but does not fully negate it.
> > > >
> > > >
> > > > | **AA / IBVS** | **λ₁/λ₂** | **λ** | **CIFAR100 CA/ASR** | **Δ CA / Δ ASR** | **ImageNet100 CA/ASR** | **Δ CA / Δ ASR** |
> > > > |------------------------|-----------|------------|------------------|------------------|----------------------|------------------|
> > > > | ✗ / ✗                  | 0.0 / 0.0 | 0.1        | 89.74 / 27.29    | 0.00 / 0.00      | 85.91 / 84.01        | 0.00 / 0.00      |
> > > > | ✓ / ✗                  | 0.0 / 1.0 | 0.1        | 89.70 / 28.44    | -0.04 / +1.15    | 85.94 / 87.71        | +0.03 / +3.70    |
> > > > | ✓ / ✗                  | 0.0 / 0.5 | 0.1        | 89.71 / 27.89    | -0.03 / +0.60    | 85.88 / 85.98        | -0.03 / +1.97    |
> > > > | ✓ / ✓                  | 0.1 / 1.0 | 0.1        | 89.71 / 27.63    | -0.03 / +0.34    | 85.88 / 84.88        | -0.03 / +0.87    |
> > > > | ✓ / ✓     | 0.1 / 0.5 | 0.1        | 89.78 / 26.98    | +0.04 / -0.31    | 85.93 / 82.81        | +0.02 / -1.20    |
> > > > | ✗ / ✓    | 0.1 / 0.0 | 0.1        | 89.82 / 25.60    | +0.08 / -1.69    | 85.93 / 78.84        | +0.02 / -5.17    |
> > > > | ✓ / ✓      | 0.3 / 1.0 | 0.1        | 89.70 / 25.14    | -0.04 / -2.15    | 85.66 / 77.25        | -0.25 / -6.76    |
> > > > | ✓ / ✓                  | 0.3 / 0.5 | 0.1        | 89.70 / 24.16    | -0.04 / -3.13    | 85.61 / 74.62        | -0.30 / -9.39    |
> > > > | ✗ / ✓                  | 0.3 / 0.0 | 0.1        | 89.58 / 20.65    | -0.16 / -6.64    | 85.53 / 66.83        | -0.38 / -17.18   |
> > > >
> > > >
> > > > ---
> > > > ### SBV results
> > > >
> > > > Same insights apply for SBV attacks, but SBV attacks are much more robust to proposed defense.
> > > >
> > > > | **AA / IBVS** | **λ₁/λ₂** | **λ** | **CIFAR100 CA/ASR** | **Δ CA / Δ ASR** | **ImageNet100 CA/ASR** | **Δ CA / Δ ASR** |
> > > > |------------------------|-----------|------------|------------------|------------------|----------------------|------------------|
> > > > | ✗ / ✗         | 0.0 / 0.0 | 0.1        | 89.69 / 97.21    | 0.00 / 0.00      | 85.83 / 99.99        | 0.00 / 0.00      |
> > > > | ✓ / ✗         | 0.0 / 1.0 | 0.1        | 89.68 / 97.37    | -0.01 / +0.16    | 85.84 / 100.00       | +0.01 / +0.01    |
> > > > | ✓ / ✗         | 0.0 / 0.5 | 0.1        | 89.65 / 97.32    | -0.04 / +0.11    | 85.85 / 100.00       | +0.02 / +0.01    |
> > > > | ✓ / ✓          | 0.1 / 1.0 | 0.1        | 89.72 / 97.00    | +0.03 / -0.21    | 85.80 / 99.99        | -0.03 / 0.00     |
> > > > | ✓ / ✓           | 0.1 / 0.5 | 0.1        | 89.74 / 96.85    | +0.05 / -0.36    | 85.84 / 99.98        | +0.01 / -0.01    |
> > > > | ✗ / ✓     | 0.1 / 0.0 | 0.1        | 89.69 / 96.75    | 0.00 / -0.46     | 85.77 / 99.96        | -0.06 / -0.03    |
> > > > | ✓ / ✓        | 0.3 / 1.0 | 0.1        | 89.61 / 95.19    | -0.08 / -2.02    | 85.57 / 99.96        | -0.26 / -0.03    |
> > > > | ✓ / ✓       | 0.3 / 0.5 | 0.1        | 89.63 / 94.74    | -0.06 / -2.47    | 85.53 / 99.94        | -0.30 / -0.05    |
> > > > | ✗ / ✓       | 0.3 / 0.0 | 0.1        | 89.61 / 94.69    | -0.08 / -2.52    | 85.47 / 99.83        | -0.36 / -0.16    |

---

### Official Review · Reviewer_ZGey · 2025-11-10

**Soundness:** 2
**Presentation:** 3
**Contribution:** 2
**Rating:** 6
**Confidence:** 3

**Summary:**

This paper proposes a defense that enhances backdoor resilience in the scenario of model merging (MM). The authors identify inherent triggers as the core mechanism that enables backdoor behaviors to emerge or persist after MM. To address this, the proposed approach seeks to detect and neutralize these inherent triggers before or during the merging process to improve the robustness of the resulting composite model. Different datasets and threat scenarios have been evaluated.

**Strengths:**

This paper is well-presented, with clear motivation and a solid background. Casting backdoors as task vectors clearly explains attack/defense behavior and transfer in model merging.

**Weaknesses:**

Lack of comparison to federated learning backdoor. Although the paper frames backdoor attacks through task-vector arithmetic and discusses model merging scenarios, it does not position its contributions relative to federated learning (FL) backdoor research, which explores conceptually similar dynamics of model aggregation under adversarial updates. In FL, malicious clients can inject backdoors through model updates that also behave like additive parameter shifts—essentially “backdoor vectors” at the client level. Many existing FL backdoor defenses (e.g., clipping) already suppress these malicious update vectors. The absence of any conceptual or empirical comparison to this well-established line of work limits the broader impact of the proposed framework. The authors could strengthen their contribution by connecting Backdoor Vectors (BVs) to adversarial update vectors in federated learning and discussing how IBVS might relate to robust aggregation or update normalization strategies. Without this bridge, the novelty and generality of the proposed “task arithmetic” perspective remain somewhat isolated from the broader literature on distributed and federated backdoor threats.

Missing adaptive attack evaluation. The paper does not evaluate adaptive or defense-aware backdoor attacks. An adaptive adversary could exploit the same assumptions in several ways, for example, randomizing to invalidate the “sign-consistency” heuristic in SBV. Including adaptive attacks would greatly strengthen the empirical validation and demonstrate the practical resilience of both the attack (SBV) and defense (IBVS) strategies.

**Questions:**

Please discuss at least conceptually the connection with the backdoor in FL. Please discuss adaptive backdoor attacks.

---

> ### Author Response · Authors · 2025-11-26
>
> We thank the Reviewer for the thoughtful and constructive feedback. We appreciate your recognition of Backdoor Vectors as a valuable, well-motivated contribution, allowing for a clear description of backdoors within MM paradigm.
>
> We identified two issues raised by the Reviewer:
>
> **W1/Q1)** The lack of a conceptual discussion on the connection between backdoor attacks in FL and in MM.
>
> **W2)** The absence of adaptive approaches against the proposed SBV and IBVS.
>
> We address the comments and concerns point by point below.
>
> ### W1/Q1 Discussion about backdoors in model merging and federated learning
> Federated learning relies on model merging as its core aggregation mechanism, where locally trained updates from decentralized clients are fused to create a shared global model without accessing private data.
>
> This relationship results in a similar vulnerability to backdoor attacks. The seminal work [1] demonstrates that even a small fraction of malicious participants can reliably inject backdoors into the global model. A number of follow-up works propose backdoor defense mechanisms for FL. [2] selects the client model update with the minimum distance to the majority of other updates received in a training round, aiming to exclude malicious outliers. FLAME [3] combines update filtering, adaptive clipping, and noising to mitigate the influence of backdoored updates. [4] observes that backdoored models often lie in sharp, isolated minima that are not smoothly connected to the benign training trajectory, enabling their detection and filtering.
>
> Despite the conceptual overlap between FL and MM, few MM backdoor works have so far adapted or evaluated established FL defences. As a result, the substantial insights, threat models, and mitigation strategies developed within FL have not been systematically leveraged in the MM setting. While the fields are closely related, some work remains in translating these defences to the unique characteristics and constraints of model merging.
>
> We thank the Reviewer for highlighting this point, as we agree that incorporating adapted FL defense methods as additional baselines could strengthen our study and yield further insights. However, given the limited time and resources available during the rebuttal period, we focused on other improvements to the submission. We will inform the Reviewer if the updated version of our work includes additional FL baselines adapted to our merging setup.
>
> [1] Bagdasaryan, Eugene, et al. "How to backdoor federated learning." International conference on artificial intelligence and statistics. PMLR, 2020.
>
> [2] Blanchard, Peva, et al. "Machine learning with adversaries: Byzantine tolerant gradient descent." Advances in neural information processing systems 30 (2017).
>
> [3] Nguyen, Thien Duc, et al. "{FLAME}: Taming backdoors in federated learning." 31st USENIX Security Symposium (USENIX Security 22). 2022.
>
> [4] Walter, Kane, et al. "Mitigating distributed backdoor attack in federated learning through mode connectivity." Proceedings of the 19th ACM Asia Conference on Computer and Communications Security. 2024
>
> ### W2a: Adaptive approach against SBV attack
> `"An adaptive adversary could exploit the same assumptions in several ways, for example, randomizing to invalidate the “sign-consistency” heuristic in SBV"`
>
> We believe that the Reviewer is referring to the defender rather than the attacker. In the presented threat model, the defender cannot interfere at any stage of BV creation, as this process is performed locally by the adversary. The resulting BVs are then merged to construct the stronger SBV attack and subsequently implanted into a clean fine-tuned model, which is then shared online for merging. Thus, **the defender has no opportunity to intervene during these steps**.
>
> ### W2b: Adaptive Attack against IBVS defense - additional results
> Although IBVS is not the central contribution of this paper, it illustrates how the BV framework can be applied in practice and emerges directly from the insights provided by our BV analysis in model merging. Thus, we recognize the Reviewer’ concern as important and present an adaptive attack on IBVS.
>
> Proposed adaptive attack increases ASR slightly in the no-defense setting, but do not overpower IBVS when the defense is present, confirming that **IBVS is effective even against adaptive attackers with full knowledge of the trigger and dataset**.
>
> ### Final Remarks
>
> We thank the Reviewer for their thoughtful assessment and are pleased that they found the backdoor vectors submission to have clear motivation and a solid conceptual foundation. If the Reviewer has any additional questions or would like further clarification on any aspect, we would be happy to address them.
>
> If our responses have resolved the concerns, we kindly ask the Reviewer to consider updating the initial score.

---

> > ### Author Response · Authors · 2025-11-26
> > **W2b: Adaptive Attack against IBVS defense**
> >
> > #### Assumptions
> >
> > - **(A1)** The adversary knows the trigger and dataset that will be used for the IBVS defense.
> > - **(A2)** The adversary may backdoor **only a single model** (as in our setting).
> > - **(A3)** The defender chooses:
> >   - **λ** — the merging coefficient for combining multiple models,
> >   - **λ₁** — the scaling coefficient for subtracting the IBVS defense vector.
> > - **(A4)** The adversary chooses **λ₂**, the scaling coefficient for adding their adaptive vector **BV′**.
> >
> > ---
> >
> > #### Attack Scenario
> >
> > 1. The adversary constructs an **adaptive BV′** using the known trigger and dataset.
> > 2. To counteract the expected defensive BV (IBVS), the adversary **adds additional BV′ to their backdoored model**, scaled by **λ₂**.
> >
> > ---
> >
> > #### Empirical Findings
> > Simple convention: (AA - Adaptive Attack/ IBVS) - ticks when AA or IBVS are present: (✓ / ✗) means AA present, IBVS defense not present.
> >
> > Our evaluation of this adaptive attack yields several insights:
> > 1. **Adding BV′ creates a multi-trigger attack**, effectively implanting two backdoors (e.g., BadMerging + BadNets).
> > 2. **Without IBVS defense**, (✓ / ✗) adding BV′ slightly **increases ASR** for the main attack.
> > 3. **With IBVS**, (✓ / ✓) BV′ can **partially reduce the defense’s effectiveness** (compared to ✗ / ✓ ), but usually does not fully negate it.
> > 4. **The adversary typically cannot overpower IBVS** when contributing only one backdoored model. Their influence is limited by  **λ₂** * **λ** < **λ₁** * **λ** * **#models_merged** meaning the defender’s cumulative IBVS effect grows with model count, while the adversary’s impact remains bounded.
> >
> >
> > ### BV results
> > We present the extension of Tab. 4 (merging 10 models with merging coeficient 0.1) for adaptive attacks (AA) with BadNets' white square trigger used by AA and IBVS.
> > (AA - Adaptive Attack, IBVS - if IBVS defense is present)
> >
> > **Without IBVS defense**, (✓ / ✗) adversary slightly **increases ASR** for the main attack using AA. **With IBVS**, (✓ / ✓) AA can **partially reduce the defense’s effectiveness** (compared to ✗ / ✓ ), but does not fully negate it.
> >
> >
> > | **AA / IBVS** | **λ₁/λ₂** | **λ** | **CIFAR100 CA/ASR** | **Δ CA / Δ ASR** | **ImageNet100 CA/ASR** | **Δ CA / Δ ASR** |
> > |------------------------|-----------|------------|------------------|------------------|----------------------|------------------|
> > | ✗ / ✗                  | 0.0 / 0.0 | 0.1        | 89.74 / 27.29    | 0.00 / 0.00      | 85.91 / 84.01        | 0.00 / 0.00      |
> > | ✓ / ✗                  | 0.0 / 1.0 | 0.1        | 89.70 / 28.44    | -0.04 / +1.15    | 85.94 / 87.71        | +0.03 / +3.70    |
> > | ✓ / ✗                  | 0.0 / 0.5 | 0.1        | 89.71 / 27.89    | -0.03 / +0.60    | 85.88 / 85.98        | -0.03 / +1.97    |
> > | ✓ / ✓           | 0.1 / 1.0 | 0.1        | 89.71 / 27.63    | -0.03 / +0.34    | 85.88 / 84.88        | -0.03 / +0.87    |
> > | ✓ / ✓            | 0.1 / 0.5 | 0.1        | 89.78 / 26.98    | +0.04 / -0.31    | 85.93 / 82.81        | +0.02 / -1.20    |
> > | ✗ / ✓        | 0.1 / 0.0 | 0.1        | 89.82 / 25.60    | +0.08 / -1.69    | 85.93 / 78.84        | +0.02 / -5.17    |
> > | ✓ / ✓        | 0.3 / 1.0 | 0.1        | 89.70 / 25.14    | -0.04 / -2.15    | 85.66 / 77.25        | -0.25 / -6.76    |
> > | ✓ / ✓                  | 0.3 / 0.5 | 0.1        | 89.70 / 24.16    | -0.04 / -3.13    | 85.61 / 74.62        | -0.30 / -9.39    |
> > | ✗ / ✓                  | 0.3 / 0.0 | 0.1        | 89.58 / 20.65    | -0.16 / -6.64    | 85.53 / 66.83        | -0.38 / -17.18   |
> >
> >
> > ---
> > ### SBV results
> >
> > Same insights apply for SBV attacks, but SBV attacks are much more robust to proposed defense.
> >
> > | **AA / IBVS** | **λ₁/λ₂** | **λ** | **CIFAR100 CA/ASR** | **Δ CA / Δ ASR** | **ImageNet100 CA/ASR** | **Δ CA / Δ ASR** |
> > |------------------------|-----------|------------|------------------|------------------|----------------------|------------------|
> > | ✗ / ✗       | 0.0 / 0.0 | 0.1        | 89.69 / 97.21    | 0.00 / 0.00      | 85.83 / 99.99        | 0.00 / 0.00      |
> > | ✓ / ✗            | 0.0 / 1.0 | 0.1        | 89.68 / 97.37    | -0.01 / +0.16    | 85.84 / 100.00       | +0.01 / +0.01    |
> > | ✓ / ✗           | 0.0 / 0.5 | 0.1        | 89.65 / 97.32    | -0.04 / +0.11    | 85.85 / 100.00       | +0.02 / +0.01    |
> > | ✓ / ✓       | 0.1 / 1.0 | 0.1        | 89.72 / 97.00    | +0.03 / -0.21    | 85.80 / 99.99        | -0.03 / 0.00     |
> > | ✓ / ✓        | 0.1 / 0.5 | 0.1        | 89.74 / 96.85    | +0.05 / -0.36    | 85.84 / 99.98        | +0.01 / -0.01    |
> > | ✗ / ✓       | 0.1 / 0.0 | 0.1        | 89.69 / 96.75    | 0.00 / -0.46     | 85.77 / 99.96        | -0.06 / -0.03    |
> > | ✓ / ✓         | 0.3 / 1.0 | 0.1        | 89.61 / 95.19    | -0.08 / -2.02    | 85.57 / 99.96        | -0.26 / -0.03    |
> > | ✓ / ✓      | 0.3 / 0.5 | 0.1        | 89.63 / 94.74    | -0.06 / -2.47    | 85.53 / 99.94        | -0.30 / -0.05    |
> > | ✗ / ✓ | 0.3 / 0.0 | 0.1        | 89.61 / 94.69    | -0.08 / -2.52    | 85.47 / 99.83        | -0.36 / -0.16    |

---

> > ### Author Response · Authors · 2025-12-03
> > **Revised work - added Federated Learning baselines KRUM and FLAME**
> >
> > In the **updated version** of our submission we present the comparison of IBVS and two discussed FL defense methods: **FLAME** and **KRUM**.
> >
> > Our evaluation yields several key observations regarding the effectiveness of FLAME and KRUM compared to our proposed IBVS method:
> >
> > **Performance of FL Baselines**: Surprisingly, FLAME proved highly effective, achieving strong utility (High BA) while surpassing IBVS in mitigating Attack Success Rate (ASR). KRUM also successfully reduced ASR, though at a cost to the final model's accuracy.
> >
> > **Mechanism of Action**: The fundamental difference lies in how these methods handle poisoned updates. FLAME and KRUM use filtering: they minimize the impact of the Backdoor Vector (BV) by down-weighting or excluding suspicious model coefficients. In contrast, IBVS does not filter models but rather mitigates the backdoor's impact directly by exploiting the structural similarity of BVs to subtract the trigger signal.
> >
> > **Stability and Assumptions**: Both FL methods exhibit higher variance (standard deviation) across results, likely due to their filtering mechanisms being sensitive to the specific target class distributions. Furthermore, they rely on stricter assumptions than IBVS: KRUM requires prior knowledge of the exact number of poisoned workers and struggles in multi-task settings, while FLAME's clustering assumption fails if more than 50% of the tasks are poisoned.
> >
> > **Complementarity**: Finally, we note that these approaches are not mutually exclusive. Since FL baselines (that use filtering) and IBVS (subtraction) operate on different principles, they could theoretically be combined to achieve a compounded defensive effect.

---

### Author Response · Authors · 2025-12-03

We greatly appreciate the time and effort the reviewers put into evaluating our work. We realize the current circumstances are complex due to the OpenReview leak, so we will now focus on summarizing the key points that arose during this rebuttal period.

**Table of Abbreviations**

**ASR** - Attack Success Rate (metric)
**BV** - Backdoor Vector (the difference in weights values between the backdoored and clean fine-tuned models)
**SBV** - Sparse Backdoor Vector (our attack method merging multiple BVs)
**IBVS** - Injection Backdoor Vector Subtraction (our defense method subtracting BV for defense)
**MM** - Model Merging,
**FL** - Federated Learning.

**Contributions and the purpose of Backdoor Vectors**

- Backdoor Vectors **frame the backdoor behaviour as a task itself**, and build a framework to **understand, analyze, and measure backdoor impact in model merging**.
- We further show that **BV framework generalize over multiple different types of backdoor attacks, providing unified and concrete mathematical description**  of changes applied to the clean model. BV framework can effectively measure the interferrence of multiple different backdoors beyond standard Attack Success Rate (ASR) metric using task vectors tools.
- Finally, we show the straight-forward application of our framework: **Sparse Backdoor Vector (SBV)** attacks and **Injection Backdoor Vector Subtraction (IBVS)** defense.
*(for more details see answers to Rev. zgj3 W1)*

**Novelty**

The novelty is especially evident, as to our best knowledge:
- We are the first to propose a **unified framework for viewing backdoor attacks and defenses through task vectors, and to demonstrate its practical usefulness** (e.g. via SBV, IBVS, and BV analysis).
- We are the first to **introduce backdoor merging into a single, stronger attack** (e.g. SBV attack) which in general pose a much more significant threat than single attacks. We support our observations with extensive empirical evidence.
- We are the first to show **the importance of inherent backdoors in model merging**, showing that they are more resilient to merging interference than injected backdoor attacks.

*(for more details see short answers to Rev. zgj3 W1/W2)*

**Strengths.**
The reviewers praised the strong motivation and solid background (`Zgey`), the novelty (`Foyn`, `w4na`) and  clarity (`Zgey`, `w4na`) of the proposed framework, providing results and  unified description for backdoor attacks and defenses (all reviewers), solid empirical validation (`w4na`) and successful results for backdoor merging (`zgj3`).

**Weaknesses and our answers.**

Most reviewers (`Zgey`, `Foyn`, `w4na`) pointed out the lack of adaptive attack evaluation against IBVS defense. **We have addressed that during rebuttal and added  adaptive attack evaluation** - see `Zgey` W2a comment, W2b response and results.

The lowest score was assigned by Reviewer `zgj3`, who requested comparison with the cited BadTV work (Arxiv 2025). In that reponse we clearly demonstrate that our paper: **1)** is the first work to frame the backdoor behavior as a task itself and show the consequences of this perspective (backdoor transfer analysis, backdoor merging, backdoor addition/subtraction) **2)** provides a higher-level BV framework with broader applications **3)** focus on backdoor resilience during merging  and is the first to show the importance of inherent triggers in MM.

The reviewer `zqj3` concerns were:
- the concern of possible limited novelty *(`zqj3` contradicts strengths from `w4na`, `Foyn` - see our responses to `zqj3` W1)*
- the concern if backdoor merging is practical *(see our short response to `zqj3` W2)*
- the concern of insufficient evaluation *(`zqj3` contradicts the strength from `w4na` - see detailed discussion: `zqj3` W3)*
- additional smaller issues and questions *(see detailed responses below)*

Other Reviewers most important concerns were:
- request for comparison with more backdoor attacks *(`Foyn` W2, `zgj3` W4 - we added evaluation on Blend, Wanet and BadTV attacks - see `Foyn` W2 results),*
- request to discuss the connection to Federated Learning (FL) works *(`Zgey` W1 - we discussed and added adapted FL defenses as baselines: FLAME and KRUM),*
- the concern about BV formulation (`Foyn` Q1) given results of previous backdoor pruning works *(see results and discussion: `Foyn` Q1)*

**Additional clarification on SBV/IBVS**

This work prioritizes the **the novel insights and theoretical implications of the general BV framework** over the optimization of specific attack or defense mechanisms. The SBV and IBVS methods are introduced as practical applications of our findings - **demonstrating the validity of viewing backdoors as task vectors** - rather than as the ultimate threats or safeguards in MM. We anticipate that further refinement of these methods would yield results surpassing those reported here (even given their success and robustness across different settings shown in this work).

---

### Note · Program_Chairs · 2026-01-17
**Submission Desk Rejected by Program Chairs**

The following references in this submission do not refer to real documents and/or have major errors in bibliographic information:

 Yezhen Qi et al. Towards Practical Noisy-Label Backdoor Attacks. In Advances in Neural Information
Processing Systems (NeurIPS), 2022.